# NEODRAGON: MOBILE VIDEO GENERATION USING DIFFUSION TRANSFORMER

**Animesh Karnewar**   **Denis Korzhenkov**   **Ioannis Lelekas**   **Noor Fathima**   **Adil Karjauv**
**Mohsen Ghafoorian**   **Amirhossein Habibian**

Qualcomm AI Research*

{karnewar,dkorzhen,ilelekas,noor,akarjauv,mghafoor,ahabibia}@qualcomm.com

Project Page: https://qualcomm-ai-research.github.io/neodragon

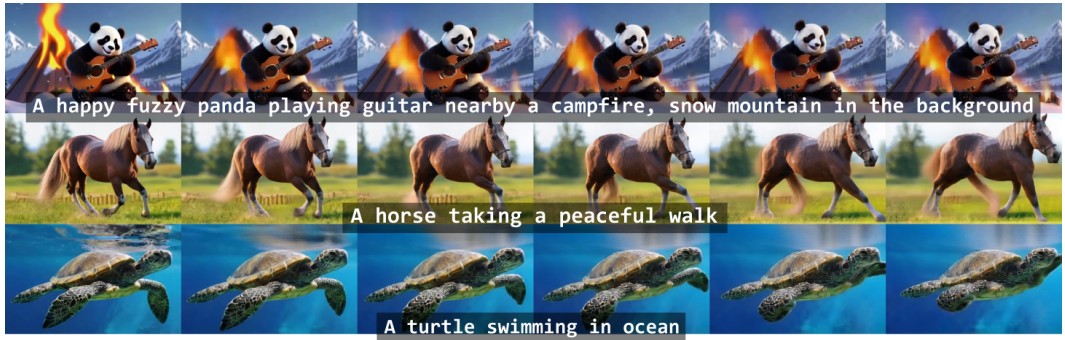

**Figure 1: Neodragon** produces high-fidelity videos with strong semantic alignment to input prompts. Shown here is a sampler of generations spanning complex motions, and both realistic and imaginative content.

## ABSTRACT

We propose Neodragon, a video DiT (Diffusion Transformer) designed to run on a low-power NPU present in devices such as phones and laptop computers. We demonstrate that, despite video diffusion transformers' huge memory and compute cost, mobile devices can run these models when carefully optimised for efficiency. To achieve this level of efficiency, i) we replace the original large Text-Encoder with a much smaller one with minimal quality loss through our novel distillation framework which doesn't require any image or video data. ii) We propose an Asymmetric Decoder distillation approach which allows us to replace the native codec-latent-VAE decoder with a more efficient one, without disturbing the generative latent-space of the video generation pipeline. iii) With our Block Pruning strategy, we remove entire blocks from the MMDiT denoiser based on their relative importance and recover original performance through a two-stage distillation process. iv) We reduce the diffusion sampling cost using our novel extended version of DMD (Distribution Matching Distillation) for the Pyramidal Flow-Matching objective. Neodragon generates 49 frames of [640×1024] resolution within ~**6.7 seconds** on the Qualcomm Hexagon NPU with a VBench total score of **81.61**, setting a new state-of-the-art for mobile video generation.

## 1 INTRODUCTION

VDMs (Video Diffusion Models) trained on internet-scale video data are poised to become the next transformative leap in artificial intelligence (Brooks et al., 2024), analogous to LLMs (Large Language Models) trained on internet-scale text corpora. However, video generation has been an active research area long before the diffusion era, predominantly driven by GAN-based approaches (Tulyakov et al., 2018; Munoz et al., 2021; Saito et al., 2020; Clark et al., 2019; Kamran et al., 2021). While GANs suffered from training instability and couldn't be scaled to larger

---

*Qualcomm AI Research is an initiative of Qualcomm Technologies, Inc.

datasets, Diffusion Models introduced a stable training objective that enabled scaling to increasingly large datasets; but at the expense of higher inference complexity (Hyvärinen, 2005; Song & Ermon, 2019; Ho et al., 2020; Song et al., 2021a;b; Karras et al., 2022; Lipman et al., 2023; Heitz et al., 2023; Liu et al., 2023). Unlike GANs, which generate samples in a single forward pass, diffusion models require numerically integrating a learned probability-flow ODE/SDE through multiple iterative steps, resulting in significantly higher computational overhead during sampling. Nonetheless, given their stability and scalability, diffusion based VDMs (Wan et al., 2025; Jin et al., 2025; Lin et al., 2024; Bar-Tal et al., 2024; Yang et al., 2025; Kong et al., 2024; HaCohen et al., 2024) were preferred over their GAN-based counterparts and are today the contemprary state-of-the-art in video generation. With Diffusion Models established as the *de facto* training paradigm for video generation, the next key question concerns the choice of model architecture. Recent systems have transitioned from spatio-temporal UNet backbones (Bar-Tal et al., 2024; Ruan et al., 2023; Ho et al., 2022a) to Transformer based designs (Zhang et al., 2025c; HaCohen et al., 2024; Cheng et al., 2024), with DiTs (Diffusion Transformers) (Peebles & Xie, 2023) emerging as the state of the art due to their superior scalability, temporal coherence, and visual fidelity (Melnik et al., 2024; Thor, 2024).

Although many frontier open-source VDMs exist today (Kong et al., 2024; Wan et al., 2025; HaCohen et al., 2024; Yang et al., 2025; Lin et al., 2024), their heavy computational demands limit accessibility; forcing reliance on cloud infrastructure with latency, privacy, and cost concerns. This causes barriers for creators in low-connectivity or resource-constrained settings. Enabling on-device generation would democratise access by allowing high-quality video synthesis directly on mobile hardware. Mobile video generation is especially important for privacy-preserving, always-available creative workflows and equitable access in bandwidth or cost-constrained regions; yet it is technically challenging due to tight compute, memory, bandwidth, and thermal/power budgets on handheld devices. To bridge this gap, we propose a set of simple yet effective techniques that systematically reduce model and runtime complexity while maintaining fidelity, enabling practical on-device video synthesis. Motivated by this vision, we introduce **Neodragon**, a DiT-based text-to-video system (see sec. 4) optimised for smartphones, delivering low-latency generation with competitive quality (see fig. 1). Our E2E (End-to-End) system runs with only a Peak-Memory usage of $\sim$**3.5GB** which is well supported by many low-power compute devices today, without affecting the other OS critical processes. We enable this mobile VDM execution through our main contributions as summarized below:

**(A)** We propose a **Text-Encoder Distillation** framework that compresses the 4.762B-parameter $T5_{\text{XXL}}$ model by 35× into a lightweight 0.130B-parameter $DT5$ (DistilT5) (Wang et al., 2025) using a newly trained 0.130B-parameter $CA$ (ContextAdapter) module, without significant quality degradation (see subsec. 3.1).

**(B)** We introduce an **Asymmetric Decoder Distillation** approach that replaces the native codec-latent-VAE decoder with a device-friendly architecture, achieving over 20× parameter reduction and enabling on-device execution with negligible quality impact (see subsec. 3.2).

**(C)** We propose a novel **MMDiT Block Pruning** strategy for the denoiser backbone, yielding >25% reduction with minimal quality loss (see subsec. 3.3).

**(D)** We extend DMD to the *Pyramidal* Flow-Matching objective (Jin et al., 2025) as our **Step Distillation** approach, reducing the number of NFEs (Neural Functional Evaluations) from 480 to 21 (>95% reduction) without affecting the VBench score (see subsec. 3.4).

## 2 RELATED WORK

We cover the most relevant related work for **on-device VDMs** here, and defer a more thorough coverage to the Appendix section B. Deploying VDMs on mobile devices is challenging due to limited compute and memory. Most mobile-optimised methods extend **UNet** architectures with aggressive compression and pruning: *AMD-Hummingbird* (Isobe et al., 2025) prunes large models via visual-feedback learning, halving parameters with minimal quality loss; *MobileVD* (Ben Yahia et al., 2024) adapts Stable Video Diffusion through spatial downscaling, multi-scale temporal representations, and structured pruning, achieving 500× efficiency gains; *SnapGen-V* (Wu et al., 2025b) combines temporal-layer NAS (Neural Architecture Search) and adversarial step distillation. *MoViE* (Karjauv et al., 2024) targets mobile video editing by introducing architectural optimisations, a lightweight autoencoder, and multi-modal classifier-free guidance distillation, achieving a 3× on-device speedup. Additionally, it employs adversarial distillation to reduce sampling to a single step, enabling real-

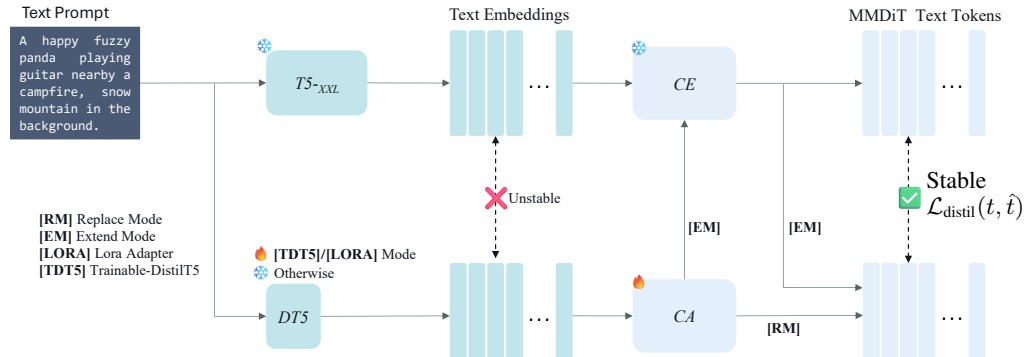

**Figure 2: Overview of the proposed Text-Encoder Distillation framework**. The original large-scale text-encoder $T5_{\text{XXL}}$ is distilled into a light-weight model via a trainable $CA$ (ContextAdapter) module, using a combination of MSE and Cosine Distance loss to align the embeddings. Multiple modes are supported in our framework – Replace Mode **[RM]**: where the new $CA$ *replaces* the original $CE$ (ContextEmbedder); Extend Mode **[EM]**: where the new $CA$ *extends* the original $CE$; Lora Mode **[LORA]**: Where the $CA$ is not a separate MLP, but LoRA Hu et al. (2022) layers on top of the $DT5$ text-encoder; and, we allow training the smaller text-encoder v/s keeping it frozen via **[TDT5]** (**Trainable-**$DT5$) mode.

time editing at 12 fps on smartphones. In contrast, **DiT**-based VDMs for on-device deployment *remain* nascent: *On-device Sora* (Kim et al., 2025) applies training-free step reduction, token merging, and dynamic loading, while *Wu et al.* (Wu et al., 2025a) introduce an extreme-compression VAE, KD-guided pruning, and adversarial step distillation to enable mobile video generation. While Mobile VDMs share a common goal, our techniques are orthogonal to these concurrent works, and their combination could enable powerful application-specific deployments.

# 3 METHOD

## 3.1 TEXT-ENCODER DISTILLATION

Given the efficiency constraints of our mobile VDM (appendix sec. C explains these and why Pyramidal-Flow is our choice for baseline), we sought a baseline latency and initially attempted to port Pyramidal-Flow (Jin et al., 2025) to Qualcomm Hexagon NPU regardless of the time required to generate a video; however, a major blocker arose: the $T5_{\text{XXL}}$ Text-Encoder's model footprint (4.726 B parameters) exceeds the mobile device's memory budget, We therefore boiled down our aim to a single guiding question: *is the full capacity of $T5_{\text{XXL}}$ actually necessary for high-quality text-to-video generation?* A direct operational corollary is whether a much smaller encoder can be substituted without perceptible fidelity loss. Motivated by distillation results in text-to-image and vision–language systems (Wang et al., 2025; Ma et al., 2025; Zhang et al., 2025a; Zhao et al., 2024)—which suggest large encoders are under-utilised for short, descriptive prompts—we hypothesise that text-to-*video* models impose similarly shallow semantic demands. Building on $DT5$ (DistilT5) (Wang et al., 2025), we propose a prompt-only Text-Encoder distillation framework tailored to video generation (see fig. 2).

We observe that naïvely matching $T5_{\text{XXL}}$ embeddings with DT5 proves to be unstable, so we focus on distilling only the components of text understanding from $T5_{\text{XXL}}$ that are relevant to video synthesis. To this end, we incorporate the $CE$ (ContextEmbedder) from the MMDiT denoiser, which maps $T5_{\text{XXL}}$ embeddings into multi-modal conditioning tokens for the denoiser, and we introduce a new learnable module, $CA$ (ContextAdapter), into the distillation setup. Our objective combines MSE and Cosine Distance losses between predicted and ground-truth MMDiT tokens, which we find to be stable with an optimal weighted combination found experimentally.

$$\mathcal{L}_{\text{distil}}(t, \hat{t}) := w_{\text{mse}} \left\| t - \hat{t} \right\|_2^2 + w_{\text{cd}} \left( 1 - \frac{t.\hat{t}}{|t|.|\hat{t}|} \right) \tag{1}$$

$$\text{where, } t = CE(T5_{\text{XXL}}(\texttt{prompt})) \text{ and, } \hat{t} = CA(DT5(\texttt{prompt})) \tag{2}$$

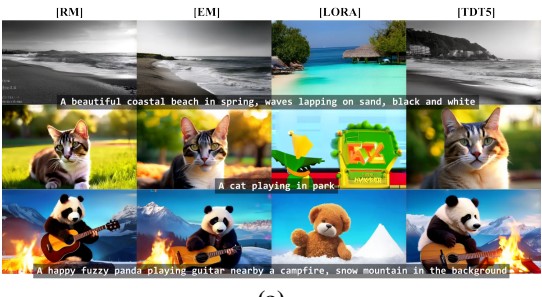

(a)

| Method | #Params (↓) (TE+CA) | VBench Score | | |
|---|---|---|---|---|
| | | Tot. (↑) | Qual. (↑) | Sem. (↑) |
| $T5_{XXL}$ Baseline | 4.732 B | 80.31 | 83.68 | 66.81 |
| $DT5$ $CA$ **[RM]** Replace | 0.260 B | 79.64 | 83.71 | 63.39 |
| $DT5$ $CA$ **[EM]** Extend | 0.266 B | 79.16 | 83.56 | 61.55 |
| $DT5$ $CA$ **[LORA]** LoRA | 0.136 B | 64.74 | 74.94 | 24.08 |
| **[TDT5]** Trainable $DT5$ | 0.136 B | 79.20 | 83.44 | 62.12 |

(b)

**Figure 3: Text-Encoder Distillation results.** (a) We visualise randomly selected frames from the generated `[49×320×512]` videos corresponding to the adjacent text prompts. (b) **#Parameters** (↓) and **VBench** (↑) scores with ContextAdapter in **[RM]**, **[EM]**, **[LORA]**, and **[TDT5]** modes.

The distillation framework supports multiple configurations, each tailored to explore a specific path over the experimental design space. The ground-truth $CE$ is a single linear layer, serving as a fixed reference throughout. In contrast, the newly introduced $CA$ is a more expressive 4-layer MLP with skip connections at every layer, designed to learn the task-specific adaptations. The framework operates in four different modes: **[RM]** Replace-Mode, where $CA$ substitutes $CE$ entirely; **[EM]** Extend-Mode, where $CA$ complements $CE$ and both of them are cascaded during inference; **[TDT5]**, which makes the $DT5$ model trainable; and **[LORA]**, which replaces the MLP-based $CA$ with LoRA (Hu et al., 2022) adapter layers on top of $DT5$. Throughout all modes, $T5_{XXL}$ and $CE$ remain frozen to provide consistent ground-truth signals for the distillation. The $CA$ is always trainable, while $DT5$ is only updated in **[TDT5]**, and in **[LORA]** mode, only the LoRA adapter layers are trainable.

The setup was trained using ∼1.4M text-prompts sourced from CommonText (Wang et al., 2025), DiffusionDB (Wang et al., 2023), a high-aesthetic subset of LAION (score > 6.5) (Schuhmann et al., 2022), and T2ICompbench (Huang et al., 2023) for 24,000 iterations with values of $w_{mse} = 1.0$ and $w_{cd} = 0.1$. Appendix section E provides more experimental details and exhaustive ablations over the loss-weights and **[LORA]** mode.

Table 3b presents a detailed quantitative evaluation, comparing multiple configurations of the $DT5$ and its associated $CA$ module. The baseline configuration, which employs the original $T5_{XXL}$ encoder, achieves a VBench Total score of 80.31, establishing the upper bound for performance within our setup. Remarkably, when $T5_{XXL}$ is replaced with the significantly smaller $DT5$ encoder paired with a 4-layer MLP-based $CA$ operating in **[RM]**, the system maintains a high VBench Total score of 79.64 (see fig. 3a). This reflects a minimal performance drop of just 0.67 percent, while delivering substantial reductions in parameter count and computational overhead.

These findings confirm our hypothesis that the full capacity of $T5_{XXL}$ is not required for high-quality video synthesis; and a carefully distilled encoder can serve as a viable drop-in replacement without compromising user experience or output fidelity. The **[RM]** configuration is integrated into the final end-to-end Neodragon pipeline (see appendix fig. H1). Notably, while the original $T5_{XXL}$ text encoder was infeasible for on-device execution, the distilled version achieves a latency of just **3ms** on Qualcomm Hexagon NPU.

## 3.2 Asymmetric Decoder Distillation

Having addressed the challenge of the large text encoder, we proceeded to port the model to the device, only to encounter a second bottleneck. Although the decoder in Pyramidal-Flow is relatively lightweight in terms of parameters (226M), its forward computation graph requires storing large 4D feature-map buffers. This made it impossible to fit even a single forward pass of the decoder (for our generation resolution) on the Qualcomm Hexagon NPU.

Rather than designing a new *mobile-friendly* codec-latent-VAE from scratch—which would require prohibitively large amounts of data and compute—we frame this challenge as another distillation problem. Existing open-source VDMs (Yang et al., 2025; Jin et al., 2025; Kong et al., 2024; Ha-Cohen et al., 2024; Wan et al., 2025; Lin et al., 2024) each employ their own codec-latent-VAEs, resulting in diverse video latent spaces for diffusion. We hypothesize that the decoder from one of these models may be sufficiently efficient to serve as a mobile-friendly candidate, which we can

**Table 1: Quantitative Evaluation of Asymmetric Decoder Distillation**. We report the DAVIS (Pont-Tuset et al., 2017) **PSNR** (↑) using the original Encoder (without modification and finetuning), using our pipeline's Encoder (after distillation), and **VBench** scores (↑) for evaluating the reconstruction performance and the generation performance respectively.

| Method | #Params (↓) Decoder | GPU Lat. (↓) | PSNR (↑) Orig. Enc. | PSNR (↑) Our Enc. | VBench Score | | |
|---|---|---|---|---|---|---|---|
| | | | | | Tot(↑) | Qual(↑) | Sem(↑) |
| PF Native Decoder | 226M | 2.496s | 29.12 | 29.12 | 80.31 | 83.68 | 66.81 |
| WAN modified | 74M | 1.666s | 31.47 | 29.18 | 80.36 | 83.82 | 66.55 |
| Cosmos CV `[8x8x8]` | 63M | 0.451s | 29.34 | 29.45 | 79.96 | 83.37 | 66.35 |
| LTXVideo modified | 237M | 1.738s | 28.34 | 29.46 | 80.34 | 83.75 | 66.68 |
| (Our) TinyAEHV modified | 10M | 0.851s | 27.71 | 28.40 | 80.25 | 83.51 | 67.19 |

distill into our pipeline. This hypothesis raises two key questions: (i) Are the video latent spaces across different models easily transferable (through lightweight finetuning)? and (ii) How can we reconcile disparities in latent compression factors among these models? To address both questions within a unified empirical framework, we propose an Asymmetric Decoder Distillation approach.

Our proposed framework comprises three components (see appendix fig. F1). **First**, we introduce asymmetry into the base model's codec-latent-VAE by retaining the original encoder, $\mathcal{E}_{\text{enc}}$, to produce latents $z = \mathcal{E}_{\text{enc}}(x)$, while replacing the original decoder $\mathcal{E}_{\text{dec}}$ with a new one, $\mathcal{F}_{\text{dec}}$, to reconstruct videos as $\hat{x} = \mathcal{F}_{\text{dec}}(z)$. Since $\mathcal{F}_{\text{dec}}$ was originally trained for a different latent space, fine-tuning is essential. However, before fine-tuning, we must resolve the mismatch in compression factors between the native encoder and the asymmetric decoder. **Second**, we minimally adapt the decoder architecture to match the fixed encoder's compression factor of `[8×8×8]`. This adjustment involves either adding or removing blocks, depending on the decoder's original compression ratio. When new blocks are introduced, we reuse the existing architectural design as much as possible and minimise additional parameters. Finally **third**, we fine-tune the entire setup end-to-end using a reconstruction objective, $\mathcal{L}(x, \hat{x})$, combining MSE and LPIPS losses (Zhang et al., 2018). The encoder remains frozen to preserve the latent space required by the MMDiT, which also allows us to omit the KL regularizer typically employed in VAE training.

We train this setup using a dataset of ∼350K videos (and corresponding prompts) sourced as: ∼253K videos from the Stock Videos subset (Mixkit, Pexels, Pixabay) of OpenSora (Lin et al., 2024), and ∼87K from Panda-70M (Chen et al., 2024). The training is done for 200,000 iterations on eight 80GB Nvidia-H100 GPUs. Appendix section F provides details about modifications applied to each decoder, and more details about the training setup.

Table 1 summarises our experiments with the proposed Asymmetric Decoder Distillation framework. Remarkably, even with minimal architectural modifications, all decoder variants perform well. The PSNR scores on the DAVIS (Pont-Tuset et al., 2017) test set average above **29 dB**, indicating that the asymmetric latent VAE can faithfully reconstruct video signals while operating through the frozen generative latent space of MMDiT. Although these experiments are preliminary, they provide strong empirical evidence for the universal nature of compressive video latent spaces learned by different models, demonstrating that such spaces can be transferred between each other with relatively low fine-tuning cost. For our deployment, the TinyAEHV decoder (Boer Bohan, 2025) proved to be the most parameter-efficient and mobile-friendly option, because of its design and portability; even though other decoders demonstrate better performance. While the native decoder could not run on the Qualcomm Hexagon NPU, the modified version achieves a latency of **143 ms** when decoding a `[49×320×512]` video from a latent tensor of shape `[7×40×64]`. This distilled decoder is integrated into our final optimised Neodragon pipeline (see appendix fig. H1).

## 3.3 MMDiT Block Pruning

After addressing the two major challenges of the oversized Text-Encoder and the unoptimised decoder, we successfully obtained our first end-to-end Qualcomm Hexagon NPU latency measurement of ∼**184.2s**. While this result demonstrates the feasibility of mobile video generation, the total runtime of approximately three minutes to produce a 2-second video at a relatively small resolution of `[320×512]` is far from our goal. Interestingly, the Text-Encoder and the Decoder, which initially prevented on-device execution, now account for only 0.2s of the total latency. The remaining 184s are required for the *spatio-temporally pyramidal and causal* latent generation performed by the MMDiT denoiser. This observation motivated our next two optimisation directions: (i) reducing

**Table 2: Quantitative Evaluation of MMDiT Block Pruning**. Performance of MMDiT Block-Pruning across model sizes, reported via VBench scores (↑) after Stage-1/2 fine-tuning, with model size (#Params, ↓) and Qualcomm Hexagon NPU latency (↓). Latency reported only for base and selected 18-block variant used in the final pipeline to minimise profiling costs.

| Stage | Method | #Parameters (↓) | Mobile Latency (↓) | VBench Score | | |
|---|---|---|---|---|---|---|
| | | | | Tot.(↑) | Qual.(↑) | Sem.(↑) |
| Original model | 24 Blocks MMDiT | 2.028 B | 1.15s | 80.31 | 83.68 | 66.81 |
| **1** | 22 Blocks MMDiT | 1.858 B | - | 79.82 | 83.30 | 65.92 |
| | 20 Blocks MMDiT | 1.688 B | - | 78.65 | 82.36 | 63.82 |
| | 18 Blocks MMDiT | 1.518 B | 0.74s | 78.39 | 81.58 | 65.63 |
| | 16 Blocks MMDiT | 1.348 B | - | 74.59 | 78.74 | 57.99 |
| **2** | 18 Blocks MMDiT | 1.518 B | 0.74s | 80.21 | 83.54 | 66.90 |
| | 16 Blocks MMDiT | 1.348 B | - | 78.62 | 82.40 | 63.50 |

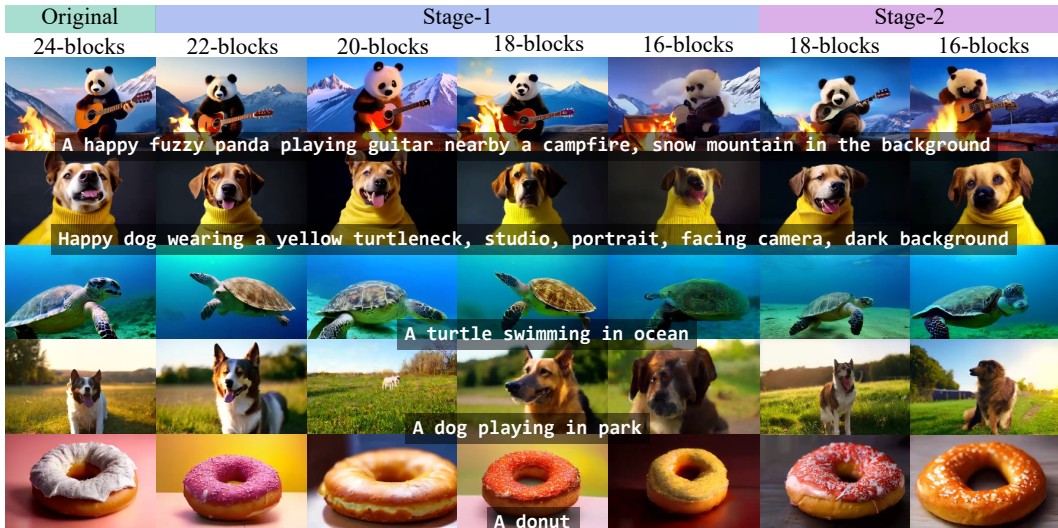

**Figure 4: Qualitative Evaluation of MMDiT Block Pruning.** We visualise randomly selected frames from the generated [49×320×512] videos, across models with different pruned-sizes, after Stage-1 finetuning, and Stage-2 finetuning.

the size of MMDiT in order to accelerate each denoising step; and (ii) reducing the number of denoising steps required to generate the latent video. In this subsection, we focus on the first direction, while the second direction is explored in the next. To this end, we propose a block-pruning strategy inspired by SANA-1.5 (Xie et al., 2025), but adapted to the MMDiT architecture and extended with a Full Teacher fine-tuning stage.

**Analysing Block Importance.** The MMDiT denoiser, denoted as $\mathcal{D}$, can be expressed as a composition of $N$ blocks operating on tokens concatenated from three sources: visual latent tokens $z$ (diffused with noise), textual tokens $\hat{t}$ derived from the `prompt` (see eq. 2), and the clip embedding token $\hat{c} := CLIP(\text{prompt})$. Pyramidal-Flow trains the denoiser starting from SD3.5 (Esser et al., 2024) MMDiT checkpoint, hence here $N = 24$. Due to the multi-modal nature of the MMDiT architecture, we obtain separate block importance scores for the visual and the textual tokens, represented by $BI_k^v$ and $BI_k^t$ respectively, for the $k^{\text{th}}$ block.

$$\left(z_{k+1}, \hat{t}_{k+1}\right) = \mathcal{D}_k(z_k, \hat{t}_k, \hat{c}) \tag{3}$$

$$BI_k^v := 1 - \mathbb{E}\left[\frac{z_k \cdot z_{k+1}}{\|z_k\|_2 \|z_{k+1}\|_2}\right] \text{ and } BI_k^t := 1 - \mathbb{E}\left[\frac{\hat{t}_k \cdot \hat{t}_{k+1}}{\|\hat{t}_k\|_2 \|\hat{t}_{k+1}\|_2}\right] \tag{4}$$

$$BI_k := \left(BI_k^v, BI_k^t\right) \tag{5}$$

We compute block-importance scores for each $k^{\text{th}}$ block in the MMDiT (see eq. 5), defined as the Cosine Distance between the block $\mathcal{D}_k$'s input and output tokens. To estimate these scores, we use a small but diverse calibration set of 100 text prompts, generating five sample videos for each. During

**Table 3: Quantitative Evaluation of Step Distillation.** Comparison of step-distillation methods adapted to Pyramidal Flow-Matching for video generation under two inference schedules (4-4-4, 1-1-1). The Pyramidal-Flow default configuration is to generate the first frame with 20 steps and rest with 10 at each of the three resolutions. Metrics: VBench—Total (Tot↑), Quality (Qual↑), Semantic (Sem↑), and Dynamic-Degree (DD↑).

| Method | Inference Schedule | Distillation Schedule | NFE's (↓) | Mobile Latency (↓) | VBench Score | | | |
|---|---|---|---|---|---|---|---|---|
| | | | | | Tot(↑) | Qual(↑) | Sem(↑) | DD(↑) |
| Pyramidal-Flow Native (CFG present) | (20)10×3 | N/A | 480 | 118.40s | 80.31 | 83.68 | 66.81 | 64.72 |
| | 4-4-4 | N/A | 168 | 41.44s | 75.82 | 79.90 | 59.49 | 49.17 |
| | 1-1-1 | N/A | 21 | 10.36s | 59.62 | 67.90 | 26.50 | 6.39 |
| Pyramidal Mean-Flows (CFG present) | 4-4-4 | N/A | 168 | 41.44s | 76.25 | 80.60 | 58.87 | 49.17 |
| | 1-1-1 | N/A | 21 | 10.36s | 63.44 | 70.89 | 33.62 | 20.00 |
| Pyramidal DMD | 4-4-4 | 1-1-1 | 84 | 20.72s | 80.37 | 85.21 | 61.01 | 86.11 |
| | 1-1-1 | 1-1-1 | 21 | 5.18s | 76.48 | 80.79 | 59.24 | 48.89 |
| Pyramidal Progressive Distillation | 4-4-4 | 4-4-4 | 84 | 20.72s | 78.22 | 82.41 | 61.46 | 52.50 |
| | 1-1-1 | 1-1-1 | 21 | 5.18s | 76.17 | 80.46 | 59.02 | 62.22 |
| Pyramidal Adversarial Distillation | 4-4-4 | 4-4-4 | 84 | 20.72s | 78.51 | 83.19 | 59.77 | 87.78 |
| | 1-1-1 | 1-1-1 | 21 | 5.18s | 77.47 | 81.74 | 60.39 | 64.17 |

this process, we probe the internal token representations of the MMDiT $\mathcal{D}$ at every denoising step for both CFG (Classifier-Free Guidance (Ho & Salimans, 2021)) passes—one with descriptive prompt and one with the negative. Interestingly, due to the model's multi-modal nature, the visual and textual importance of a block are not correlated (see appendix sec. G for more details). Therefore, we also assess the impact of removing each block on the final generation quality visually. Based on both the importance scores and visual impact, we select the blocks for pruning.

**Stage-1 Finetuning.** After removing the selected blocks from the MMDiT, we finetune the pruned model using ground-truth data. Specifically, we train the pruned model with our curated ~350K dataset (see subsec. 3.2) with the original Flow-Matching objective (Lipman et al., 2023). We adopt the default setting of Pyramidal-Flow, where only the current frame is denoised while conditioning on past frames sampled from ground truth videos. To improve robustness to test-time generations, these history frames are corrupted with gaussian noise during training. Surprisingly we find that only as little as 300 iterations using four 80GB Nvidia H100 GPUs are sufficient for the model to converge in this stage, and training longer (we tried till 3000 iterations) doesn't improve the performance further.

**Stage-2 Finetuning.** After establishing a lower bound on the performance achievable by the pruned models in Stage-1, we proceed with Stage-2 finetuning. In this stage, we incorporate feature-matching losses between the Full Teacher model and the pruned Student model. Specifically, we apply an MSE loss on visual tokens, a Cosine Distance loss on textual tokens, and Flow-Matching losses (Lipman et al., 2023) using both the Teacher model's outputs and ground-truth flow from the data as supervision. Ablations regarding the choice of these losses are detailed in Appendix section G. This stage exhibits slower convergence and requires a training for 60,000 iterations.

Table 2 and Figure 4 summarise our quantitative and qualitative results respectively. Notably, the 25% pruned model with 18 blocks achieves a VBench score of **80.21**, only 0.1 lower than the base 24-block model (**80.31**), enabling near-lossless compression of the MMDiT denoiser. And hence, we deploy this block pruned MMDiT in our final pipeline (see appendix fig. H1). With this reduction in the MMDiT model size, the latency of a single denoising pass (sum over three *pyramidal* resolutions) was reduced from **1.15s** to **0.74s** on Qualcomm Hexagon NPU.

We found it remarkably intriguing that Stage-2 finetuning does not work well when applied directly to the pruned model, despite including the data based Flow-Matching loss in Stage-2. We also experimented with various annealed weighting schemes during training, but none matched the performance achieved by the curriculum approach of Stage-1 followed by Stage-2. We attribute this behaviour to the optimisation landscape induced by pruning, though a deeper investigation could provide valuable insights, which we leave as an interesting direction for future work.

### 3.4 STEP DISTILLATION

Having pruned approximately 25% of the MMDiT denoiser's parameters, the video generation latency on the Qualcomm Hexagon NPU decreased from **184.2s** to **118.6s**, yielding a saving of **65.8s** and significantly improving time-to-video. However this is still a bit away from a practically de-

ployable system. Therefore, we aim to reduce latency further to make the system more practical and user-friendly, with real-time generation remaining the ultimate goal. Now with the block-pruned MMDiT, the iterative latent denoising accounts for **118.4s** of the total latency, requiring 480 NFEs (denoising steps). To accelerate generation by distilling the denoising steps, we propose a pyramidal version of DMD (Yin et al., 2024). Given space constraints, we defer the details of the rest of our adaptions to the Section H of the appendix.

Pyramidal-Flow decomposes the probability flow into $S$ stages, where the $i^{\text{th}}$ stage operates at $2^i \times$ smaller resolution than the original, where $i \in \{0, \dots, S-1\}$. Let $\text{Down}(\cdot, s)$ and $\text{Up}(\cdot, s)$ denote spatial downsampling and upsampling by a factor $s$, respectively. Each stage is parameterised by a pair of noise levels $(\sigma^i_{\text{start}}, \sigma^i_{\text{end}})$ with $1 > \sigma^i_{\text{start}} > \sigma^i_{\text{end}} > 0$, and operates on latents at resolution $\text{Down}(\boldsymbol{z}, 2^i)$. The start and end distributions for stage $i$ are defined as

$$\tilde{\boldsymbol{z}}_{\sigma^i_{\text{start}}} := (1 - \sigma^i_{\text{start}})\, \text{Up}\big(\text{Down}(\boldsymbol{z}, 2^{i+1}), 2\big) + \sigma^i_{\text{start}}\,\epsilon, \tag{6}$$

$$\tilde{\boldsymbol{z}}_{\sigma^i_{\text{end}}} := (1 - \sigma^i_{\text{end}})\, \text{Down}(\boldsymbol{z}, 2^i) + \sigma^i_{\text{end}}\,\epsilon. \tag{7}$$

A different local noise-level $\sigma^i_{\text{local}} \sim \mathbb{U}(0,1)$ is used to learn the Flow-Matching model at the $i^{\text{th}}$ stage, and the global noise-level $\sigma$ relates to the local noise-level $\sigma^i_{\text{local}}$ as $\sigma = (1 - \sigma^i_{\text{local}})\sigma^i_{\text{end}} + \sigma^i_{\text{local}}\sigma^i_{\text{start}}$. Thus, the overall Pyramidal Flow Matching objective is an aggregate over the stagewise objectives: $\mathcal{L}_{\text{pyr-FM}} := \sum_{i=0}^{S-1} \mathcal{L}^i_{\text{FM}}$. Appendix section D provides more details.

For the *Pyramidal*-DMD, at $i^{\text{th}}$ stage input $\tilde{\boldsymbol{z}}_\sigma = (1 - \sigma^i_{\text{local}})\tilde{\boldsymbol{z}}_{\sigma^i_{\text{end}}} + \sigma^i_{\text{local}}\tilde{\boldsymbol{z}}_{\sigma^i_{\text{start}}}$, the to-be-learned Student model $\mathcal{D}_\theta$ aims to predict the clean latent, parametrized as a single-step Euler solver $\tilde{\boldsymbol{z}}_\theta := \tilde{\boldsymbol{z}}_\sigma - (\sigma/(\sigma^i_{\text{start}} - \sigma^i_{\text{end}}))\mathcal{D}_\theta(\tilde{\boldsymbol{z}}_\sigma, \sigma)$, since the teacher Score-Model $\mathcal{D}$ had been trained to approximate the flow defined as a derivative w.r.t. $\sigma^i_{\text{local}}$. The so-called Fake-Score-Model $\mathcal{D}_\varphi$ is trained with pyramidal Flow Matching objective $\mathcal{L}_{\text{pyr-FM}}$ but on the distribution of student-predicted clean

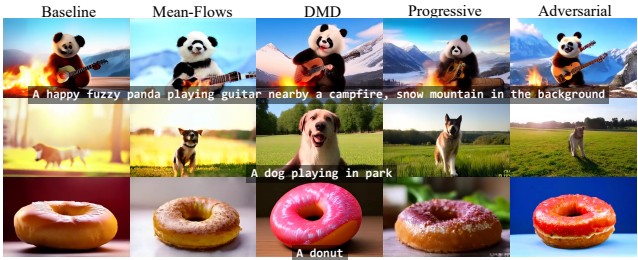

**Figure 5: Qualitative evaluation of step distillation.** We visualise randomly selected frames from the generated [49×320×512] videos, across different step distillation application on the block-pruned model for 4-4-4 configuration.

latents instead of ground-truth video latents. Having the fake model, the student network is updated with DMD loss defined through its gradient $\nabla_\theta L^i_{\text{DMD}} \propto (\mathcal{D}(\tilde{\boldsymbol{z}}_\tau, \tau) - \mathcal{D}_\varphi(\tilde{\boldsymbol{z}}_\tau, \tau)) \cdot \nabla_\theta \tilde{\boldsymbol{z}}_\theta$. The input of teacher and fake model $\tilde{\boldsymbol{z}}_\tau$ is defined as a stage-wise noisy version of student-predicted clean latent, similar to eq. 6 and eq. 7,

$$\tilde{y}_{\sigma^i_{\text{start}}} := (1 - \sigma^i_{\text{start}})\, \text{Up}\big(\text{Down}(\tilde{\boldsymbol{z}}_\theta, 2), 2\big) + \sigma^i_{\text{start}}\,\varepsilon, \tag{8}$$

$$\tilde{y}_{\sigma^i_{\text{end}}} := (1 - \sigma^i_{\text{end}})\, \tilde{\boldsymbol{z}}_\theta + \sigma^i_{\text{end}}\,\varepsilon, \tag{9}$$

$$\tilde{\boldsymbol{z}}_\tau := (1 - \tau^i_{\text{local}})\tilde{y}_{\sigma^i_{\text{end}}} + \tau^i_{\text{local}}\tilde{y}_{\sigma^i_{\text{start}}}, \tag{10}$$

where $\varepsilon \sim \mathcal{N}(0, \mathbb{I})$ and $\tau = (1 - \tau^i_{\text{local}})\sigma^i_{\text{end}} + \tau^i_{\text{local}}\sigma^i_{\text{start}}$. We follow Yin et al. (2024) and define the sample-specific weight of DMD loss as $\left\| \mathcal{D}(\tilde{\boldsymbol{z}}_\tau, \tau) - \left( \tilde{y}_{\sigma^i_{\text{start}}} - \tilde{y}_{\sigma^i_{\text{end}}} \right) \right\|_1^{-1}$. Therefore, the sample gets higher weight, if teacher model is capable to estimate its conditional flow with a smaller error. In addition we found the supervised Cauchy loss $\mathcal{L}_{\text{teacher}} = \log\left(1 + \left\| \tilde{\boldsymbol{z}}_\theta - \text{Down}(\boldsymbol{z}, 2^i) \right\|_2^2\right)$ useful for visual quality and used it with weight 0.5. During training, we update the student and fake model in alternate manner: one update of $\theta$ per two updates of $\varphi$. For student's updates we limit the set of local noise levels $\tau^i_{\text{local}}$ to four evenly selected values, and for fake model it is sampled from $\mathbb{U}(0,1)$. To obtain teacher's prediction we employ classifier-free guidance with the same hyperparameters as those recommended for Pyramidal-Flow.

Table 3 and Figure 5 summarise our quantitative and qualitative results respectively. Since the pyramidal setting of the base model operates on 3 different resolutions at the time of generation, we specifically target two different configurations, namely 4-4-4 and 1-1-1, where the denoiser spends 4

**Table 4: Comparison with state-of-the-art video generation models**. All **VBench** scores from the compared methods are extracted from their reported numbers, except for 'Wan2.1*', and 'Pyramidal-Flow$^{\downarrow}$' which are our reproduction of the scores using the same evaluation pipelines and parameters for the generated video resolution of [49×320×512] as we have used for our optimisations.

| Platform | Model | Tot.(↑) | Qua.(↑) | Sem.(↑) | Flick.(↑) | Aes.(↑) | Ima.(↑) | Obj.(↑) | Scene(↑) | Cons.(↑) |
|---|---|---|---|---|---|---|---|---|---|---|
| **Server** | Pyramidal-Flow | 81.72 | 84.74 | 69.62 | 99.49 | 63.26 | 65.01 | 86.67 | 43.20 | 26.23 |
| | Wan2.1 1.3B | 83.31 | 85.23 | 75.65 | 99.55 | 65.46 | 67.01 | 88.81 | 41.96 | 25.50 |
| | Wan2.1 1.3B* | 82.47 | 83.33 | 79.01 | 99.35 | 65.05 | 64.81 | 89.87 | 54.20 | 26.95 |
| | Open-Sora V1.2 | 79.76 | 81.35 | 73.39 | 99.53 | 56.85 | 63.34 | 82.22 | 42.44 | 26.85 |
| | CogVideoX1.5 5B | 82.01 | 82.72 | 79.17 | 98.53 | 62.07 | 65.34 | 83.42 | 53.28 | 27.42 |
| | CogVideoX 5B | 81.91 | 83.05 | 77.33 | 78.97 | 61.88 | 63.33 | 85.07 | 51.96 | 27.65 |
| | CogVideoX 2B | 81.55 | 82.48 | 77.81 | 98.85 | 61.07 | 62.37 | 86.48 | 50.04 | 27.33 |
| | Mochi-1 | 80.13 | 82.64 | 70.08 | 99.40 | 56.94 | 60.64 | 86.51 | 36.99 | 25.15 |
| | LTX-Video | 80.00 | 82.30 | 70.79 | 99.34 | 59.81 | 60.28 | 83.45 | 51.07 | 25.19 |
| | Efficient VDiT | 76.14 | - | - | 99.49 | 57.21 | - | 60.33 | - | - |
| **On-device** | Pyramidal-Flow$^{\downarrow}$ | 80.31 | 83.68 | 66.81 | 99.46 | 60.40 | 63.90 | 84.79 | 45.74 | 26.31 |
| | Snap Mobile Video DiT | 81.45 | 83.12 | 74.76 | 98.11 | 64.16 | 63.41 | 92.26 | 51.06 | 25.51 |
| | Hummingbird 16frame | 81.35 | 83.73 | 71.84 | 95.24 | 68.04 | 71.04 | 96.36 | 52.91 | 28.09 |
| | Hummingbird 26frame | 80.31 | 83.11 | 69.10 | 97.64 | 67.82 | 69.94 | 94.86 | 49.49 | 27.65 |
| | SnapGenV | 81.14 | 83.47 | 71.84 | 99.37 | 62.19 | - | 92.22 | - | 27.42 |
| | **(Ours) Neodragon** E2E | 81.61 | 83.68 | 73.36 | 99.27 | 60.71 | 59.78 | 92.37 | 56.56 | 28.09 |
| | **(Ours)** T2V Multi-Step | 80.21 | 83.54 | 66.90 | 99.40 | 59.86 | 63.56 | 83.16 | 42.54 | 26.58 |

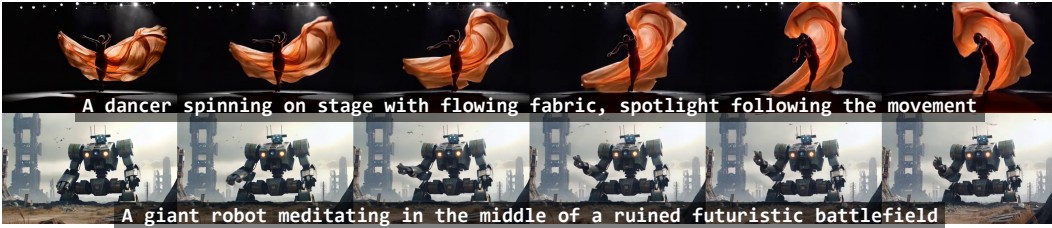

A dancer spinning on stage with flowing fabric, spotlight following the movement

A giant robot meditating in the middle of a ruined futuristic battlefield

**Figure 6: Qualitative Evaluation of E2E Neodragon pipeline.** We visualise uniformly selected frames from the [49×640×1024] videos (2x super-resolved) generated on Qualcomm Hexagon NPU.

steps and 1 step on the three denoising resolutions respectively. Except for Pyramidal Mean-Flows, all the other three adaptations provide significant VBench gains compared to the base non-distilled model's performance, especially in the lower step regime of 1-1-1. Since Pyramidal DMD yields the best VBench score of **80.37** for the 4-4-4 setting, we choose to use this step-distilled MMDiT denoiser $\mathcal{D}$, with a denoising latency of **20.72s** for the Neodragon pipeline (see appendix fig. H1).

# 4  END-TO-END MOBILE VDM

## 4.1  MODEL COMPONENT INTEGRATION

By applying step distillation, we reduced the end-to-end video generation latency to **25.96s**, bringing our system close to the threshold of interactive video generation at **2fps**. While the model maintains a strong VBench score of **80.37**, we observe noticeable visual-semantic degradation. As illustrated in Fig. 5, our proposed *Pyramidal*-DMD approach introduces colour saturation artifacts and semantic inconsistencies in the first frame. Nevertheless, the generated motion remains smooth and stable, suggesting that this issue is not fully captured by VBench metrics. We hypothesize that these semantic artifacts can be mitigated by initializing the video with a high-quality first frame generated by a separate text-to-image model. This strategy would preserve the integrity of the initial frame while leveraging our pipeline to apply coherent and descriptive motion to subsequent frames. To this end, we use SSD-1B (Gupta et al., 2024) to generate the first frame in four steps requiring 0.82s including CLIP embedding, then apply our 1-1-1 step-distilled pipeline for subsequent frames. With VAE encoding 0.94s (first frame) and QuickSRNet (Berger et al., 2023) upsampling 5ms, the full pipeline achieves **6.7s** latency (see fig. 1, and fig. 6). This configuration of ours constitutes our Mobile VDM Neodragon (see appendix fig. H1) and delivers the highest VBench score **81.61** among on-device solutions (see tab. 4). More qualitative videos are included in supplementary. Table 5 provides details of the on-device measurements done for running our Neodragon system on both a Laptop SoC (namely Snapdragon X Elite) and a Mobile SoC (namely Snapdragon 8 Elite Gen4); both powered by the Qualcomm Hexagon NPU.

**Table 5: Neodragon On-device Measurements**. We report measurements for running Neodragon on the laptop SoC Snapdragon X Elite and mobile SoC Snapdragon 8 Elite Gen4; both powered by Qualcomm Hexagon NPU. Since the VAE Encoder runs only once for the first frame, it is unoptimised. We also report peak RAM usage of each component for the laptop SoC.

| SoC / Measurement | CLIP L | CLIP G | DistilT5 | SSD1B | | VAE Enc. | VAE Dec. | MMDiT+CA | | | QuickSRNet |
|---|---|---|---|---|---|---|---|---|---|---|---|
| | | | | UNet | Dec. | | | [7×10×16] | [7×20×32] | [7×40×64] | |
| Snapdragon X Elite /Lat. ms | 5.9 | 43.6 | 3.0 | 151.5 | 378.6 | 941.7 | 143.0 | 54.9 | 101.4 | 590.2 | 4.9 |
| Snapdragon X Elite /Mem. GB | 0.49 | 2.64 | 0.03 | 2.57 | 0.17 | 0.68 | 0.21 | 3.13 | 3.15 | 3.25 | 0.01 |
| Snapdragon 8 Elite Gen4 /Lat. ms | 14.0 | 76.5 | 3.5 | 234.6 | 580.0 | 1206.5 | 248.9 | 104.7 | 218.3 | 938.3 | 6.5 |

## 4.2 MODEL COMPILATION

Deploying the model on a fixed-point NPU involves the following model compilation steps:

**Porting multi-resolution MMDiTs.** To compile static graphs, we need to port three MMDiT graphs—low, mid, and high—corresponding to the latent resolution of each stage. As illustrated in Appendix Figure D1, the PyTorch model takes past history which dynamically grows with more latent frames being included. This dynamic nature is incompatible with static compilation, hence input paddings are needed for each MMDiT graph. Specifically, we expand the last frame graph and pad zeros when running inference for earlier frames. Doing such steps requires changes to the attention mask and positional embedding implementation so that the zero paddings do not contribute to the next frame generation. This step is critical because any leakage from padded tokens can degrade temporal consistency.

**Precomputation of MMDiT inputs.** The input merge function of MMDiT's forward pass computes constant information for each stage. These include input shapes and trainable token lengths specific to the current stage. Besides, as we run a fixed amount of timesteps, we also have timestep embeddings precomputed as inputs to the MMDiT graph. Precomputing these values not only reduces runtime overhead but also ensures deterministic behaviour across different hardware backends, which is essential for reproducibility.

**Reduction of 6D tensors.** The MMDiT graph contains temporal and spatial dimensions, which makes the RoPE layer computation a 6D tensor `mul`/`add` operation. Unlike Torch dynamic graphs, high-dimensional inputs mean more complicated tiling, which usually ends up penalising performance. To simplify this, we reduce along the sin/cos and broadcast dimensions, effectively collapsing redundant axes. This enables an extreme performance boost: it reduces MMDiT-`[7×40×64]` compilation time from nearly a day to less than two hours, while improving inference latency from several seconds to sub-second. Such optimisation is particularly important for real-time applications on edge devices.

**Optimising causal mask value.** By default, the causal mask value in T5 and MMDiT is set to an extremely large number, which can cause problems during device deployment. To mitigate this, we adjust the mask value to a more suitable number—large enough to prevent the model from attending to masked tokens, but small enough to avoid overflow. This adjustment is subtle yet necessary because fixed-point arithmetic is highly sensitive to extreme constants.

**Rescaling activations in T5.** T5 includes `res_add` and `ff_add` operations with values exceeding the FP16 numerical range. To ensure numerical stability and maintain functional equivalence, we apply a scaling factor to these residual connections, effectively transforming them without altering the model behaviour. This is a common practice in low-precision deployment pipelines and is crucial for preventing saturation in activation maps, which otherwise leads to degraded generation quality.

## 5 CONCLUSION

We began with the Pyramidal-Flow pipeline and, through four optimisations, namely Text-Encoder Distillation, Asymmetric Decoder Distillation, Block Pruning, and Step Distillation, transformed it into Neodragon, an efficient on-device text-to-video generation system. Every component of the pipeline has been either refined or replaced, yet the essence of the original design persists, echoing the paradox of the **"Ship of Theseus"** (Encyclopaedia Britannica): when every part changes, does the system remain the same? Neodragon reduces latency from minutes to seconds while preserving state-of-the-art quality, marking a significant step towards practical, interactive video generation on consumer hardware. Looking ahead, achieving high-resolution and long-duration video generation on-device remains an open challenge, and enabling intelligent video-to-video editing applications offers exciting directions for future work.

## 6 ACKNOWLEDGEMENTS

This project has been a long and demanding endeavour, supported by countless individuals working diligently behind the scenes, and we take this opportunity to extend our gratitude to as many contributors as possible. We thank the Quantisation team Hanwen Xiong and Vancheeswaran Vaidyanathan for their dedicated efforts, as well as the Software team Will Zeng, Rafael Esteves, and Tushar Singhal for their invaluable work on building the demos. We are grateful to the Compute Engineering team, including Ashwath Jadhav, Prudhvi Akhil Alahari, and Ashish Chauhan. We further thank Fatih Porikli for his insightful discussions, and the supporting technical leads Frank Mayer, Ork De Rooij, and Chen Zhang for their guidance throughout the project. Our sincere appreciation goes to Michael Hoffmann (Senior Director of Engineering, Qualcomm AI Research Amsterdam) for his ongoing support, and to Lea Heusinger-Jonda for her indispensable assistance with legal approvals as well as open-sourcing and publication logistics. Last but by no means least, we acknowledge Monique Hagen for keeping the Amsterdam location running smoothly and enabling an environment where this work could thrive. Finally, we extend our thanks to all others who, in ways large or small, contributed to this project.

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

APPENDIX

## A  ADDITIONAL QUALITATIVE VIDEO SAMPLES

Please unzip the adjoining `supplementary_material.zip` and open the `index.html` file to view the supplementary video results provided to demonstrate the fidelity of our final generations. We also include qualitative video results for each of the four optimisations that we introduce to demonstrate the effectiveness of the methods that we apply.

## B  MORE RELATED WORK

**Text-to-Video Diffusion Models**. Text-to-video (T2V) generation has rapidly advanced with diffusion-based architectures, which surpass GAN-based methods in temporal consistency and scalability. Early approaches extended text-to-image diffusion models by adding temporal layers to U-Net backbones, as seen in models like *LTX-Video* (HaCohen et al., 2024) and *Open-Sora Plan* (Lin et al., 2024). However, these designs struggled with long-range temporal coherence and overall video quality. Recent models increasingly adopt transformer-based architectures for their superior ability to model global spatio-temporal dependencies. For example, *CogVideoX* (Yang et al., 2025) employs a diffusion transformer with a 3D VAE and expert transformer layers for strong text-video alignment. Similarly, *Wan* (Wan et al., 2025) adopts a large-scale transformer-based design for high-quality video synthesis, and *HunYuan Video* (Kong et al., 2024) focuses on high-fidelity generation with joint image-video training and optimised text encoders. Our work builds upon Pyramidal-Flow (Jin et al., 2025), which introduces a pyramidal flow matching strategy that progressively refines latents across spatial and temporal scales. Unlike cascaded pipelines, it unifies generation in a single diffusion transformer and supports autoregressive video generation with temporal pyramids. These inductive biases—hierarchical spatio-temporal modelling and autoregressive conditioning—make it a strong baseline for efficient, coherent video synthesis.

**Text-encoder Distillation**. Large text encoders, such as $T5_{\text{XXL}}$ or $CLIP$, are widely used in diffusion-based generative models to capture rich semantic representations. However, their size and computational cost pose significant challenges for on-device deployment and real-time generation. To address this, *DistillT5* (Wang et al., 2025) introduces a vision-guided knowledge distillation framework that compresses large T5-based encoders into smaller variants (e.g., T5-Base) while preserving semantic alignment with the visual domain. The method employs multi-stage distillation using curated datasets optimised for image quality, semantic understanding, and text rendering, achieving up to $50\times$ size reduction with minimal performance degradation. Related efforts in multimodal settings, such as *CLIP distillation* (Yang et al., 2024) and multilingual encoder distillation in *AltDiffusion* (Ye et al., 2024), further demonstrate the effectiveness of encoder compression for improving efficiency in diffusion pipelines.

Within concurrent works on mobile-optimised T2V generation, text encoder optimisation remains largely overlooked. *On-device Sora* (Kim et al., 2025) applies dynamic loading to T5 to reduce memory footprint but does not modify the encoder architecture itself. Similarly, *Wu et al.* (Wu et al., 2025a) focuses on optimising the denoising backbone and VAE components, without introducing contributions towards text encoder compression. This highlights an open research gap in systematically distilling or compressing text encoders for efficient on-device video diffusion models that we attempt to fill with our novel contribution.

**Video Decoder Optimisation**. While most research on efficiency in video diffusion models focuses on latent compression or denoising acceleration, decoder-side optimisation has received comparatively less attention. Existing works primarily explore architectural or inference-level strategies to reduce decoding overhead. *LTX-Video* (HaCohen et al., 2024) introduces a decoder that performs the final denoising step, effectively shifting part of the refinement process from the diffusion backbone to the VAE decoder, reducing the number of diffusion iterations. *WF-VAE* in Open-Sora Plan (Lin et al., 2024) proposes block-wise decoding with a *Causal Cache* mechanism to enable tiled inference for high-resolution videos under memory constraints. Similarly, *PyramidalFlow* (Jin et al., 2025) implements tile-enabled decoding and sequential offloading between CPU and GPU

to support large-scale video generation on limited hardware. Cascaded approaches such as *Imagen Video* (Ho et al., 2022b) and *Lumiere* (Bar-Tal et al., 2024) adopt multi-stage super-resolution decoders for progressive refinement, though these designs prioritise quality over on-device efficiency.

In contrast, our work introduces an *asymmetric decoder distillation* strategy that substitutes the base model's native decoder with a device-optimised architecture while preserving the original video encoding scheme. Unlike prior methods that rely on tiling or caching for memory savings, our method directly targets decoder complexity through knowledge distillation, enabling efficient deployment without altering the latent representation or retraining the diffusion backbone.

**Block Pruning**. Pruning for diffusion transformers has recently gained attention as a means to reduce inference cost without retraining from scratch. *TinyFusion* (Fang et al., 2025) introduces a learnable depth-pruning framework for DiTs, where layer masks are optimised jointly with a recoverability objective and refined through masked knowledge distillation. Similarly, *Effortless Efficiency* (Zhang et al., 2025b) proposes a model-agnostic structural pruning approach for diffusion models, learning pruning masks across the denoising process to remove redundant layers with minimal fine-tuning. For video diffusion transformers, the parallel work by *Wu et al.* (Wu et al., 2025a) adopts a sensitivity-aware tri-level pruning strategy that prunes at multiple granularities—within-layer components, attention heads, and entire blocks—guided by knowledge distillation and sensitivity analysis, as part of a broader system-level optimisation for real-time mobile generation.

In contrast, our method focuses on *block-level pruning* tailored to the MMDiT denoiser. We introduce a three-stage pipeline comprising block-importance scoring, short fine-tuning, and full teacher-model distillation. Unlike TinyFusion's differentiable depth pruning or Wu et al.'s multi-granularity sensitivity-based approach, our strategy aligns pruning units with the natural MMDiT block structure to preserve spatio-temporal attention pathways critical for video generation, while simplifying the pruning process for practical deployment.

**Step Distillation**. Reducing the number of denoising steps in diffusion models is critical for improving inference efficiency, and several step distillation strategies have been proposed. *Progressive Distillation* (Salimans & Ho, 2022) is a seminal approach that iteratively halves the number of steps by training a student to mimic the teacher's trajectory, achieving substantial speedups while preserving quality. Subsequent works explore alternative paradigms, such as *adversarial step distillation* (Zhang et al., 2024), which augments the distillation objective with adversarial losses to enhance perceptual fidelity; this strategy has been adopted in video generation pipelines such as (HaCohen et al., 2024; Wu et al., 2025a). Another interesting direction is *Distribution Matching Distillation (DMD)* (Salimans & Ho, 2022; Song et al., 2023), which aligns the student's output distribution with that of the teacher across noise levels, providing a principled framework for step reduction without progressive halving. Building on this foundation, we are the first to adapt a DMD-based step distillation method to a pyramidal flow-matching denoiser (Jin et al., 2025), enabling efficient inference while preserving the model's hierarchical spatio-temporal structure.

## C  MOBILE EFFICIENCY REQUIREMENTS

Table C1 summarises recently released text-to-video diffusion models that are publicly available. For building a mobile text-to-video system, we consider the constraints imposed by mobile hardware platforms. The primary limitations are twofold: (*i*) the total model size, which must fit within the DRAM capacity of the target device, and (*ii*) the computational complexity, as this directly influences both the power consumption and the inference latency. Regarding the first constraint, as shown in Table C1, all models except Hunyuan Video Kong et al. (2024) offer relatively compact checkpoints, providing some flexibility in model selection. However, the second constraint—compute complexity—requires a more careful analysis. In the following discussion, we explain how the two features of the model from Pyramidal-Flow, namely Causal Attention and Token-Savings (in the form of Pyramid) contribute to the mobile-friendliness of it, and thereby motivating why we select it as the foundation of our Neodragon system.

Latent video diffusion models Blattmann et al. (2023); Yang et al. (2025); Jin et al. (2025); Kong et al. (2024); HaCohen et al. (2024); Wan et al. (2025); Lin et al. (2024), building on the framework

**Table C1: Comparison of Recent Text-to-Video Models**. Rows are sorted by release date (ascending).

| Model | Release | Generation Resolution | Model-Size | VBench Total | Causal? | Token-Savings? |
|---|---|---|---|---|---|---|
| Cogvideo-X Yang et al. (2025) | 19 Sep 2024 | 49×768×1360 | 2.0B | 81.55 | ✗ | ✗ |
| Pyramidal-Flow Jin et al. (2025) | 29 Oct 2024 | 49×384×512 | 2.0B | 81.56 | ✓ | ✓ |
| Hunyuan Video Kong et al. (2024) | 3 Dec 2024 | 129×544×960 | 13.0B | 85.09 | ✗ | ✗ |
| LTXVideo HaCohen et al. (2024) | 30 Dec 2024 | 81×512×768 | 2.0B | 82.30 | ✗ | ✗ |
| Wan Wan et al. (2025) | 25 Feb 2025 | 81×480×832 | 1.3B | 84.26 | ✗ | ✗ |
| OpenSORA Lin et al. (2024) | 13 Mar 2025 | 129×768×1365 | 1.1B | 77.70 | ✗ | ✓ |

introduced by Rombach et al. Rombach et al. (2022) for images, generate videos in two stages. First, a codec-latent-VAE compresses the input from pixel space to a lower-dimensional latent space using architectures such as VQ-VAE Razavi et al. (2019) or VQ-GAN Yu et al. (2022). Second, a diffusion model is trained on these compressed latents, enabling efficient yet high-quality generation. This design balances computational efficiency with generative capacity for high-resolution video synthesis. Intuitively, the diffusion model handles the more challenging task of composing scene layout, objects, and their spatial relationships, while the VAE focuses on reconstructing textures and perceptual details. Formally, during training, an input video $x \in \mathbb{R}^{T \times H \times W \times 3}$ is mapped to a latent representation $z \in \mathbb{R}^{t \times h \times w \times c}$ via a spatio-temporal encoder: $z = \mathcal{E}_{\text{enc}}(x)$, where $t = T/f_t$ is the number of latent frames, and $h = H/f_s$, $w = W/f_s$ are the spatial dimensions reduced by a factor $f_s$ (typically 8), and $c$ are the number of latent channels. The temporal compression factor $f_t$ is usually 4 or 8, slightly different from $f_s$ (Although for Pyramidal-Flow it is also 8). During inference, the diffusion model generates a latent video $\hat{z} \in \mathbb{R}^{t \times h \times w \times c}$ starting from Gaussian noise, which is then decoded into RGB frames by: $\hat{x} = \mathcal{E}_{\text{dec}}(\hat{z})$.

As summarised in Table C1, most open-sourced video diffusion models—except for Pyramidal-Flow —employ fully bidirectional self-attention Vaswani et al. (2017) within transformers to generate or denoise the $t \times h \times w \times c$ latent representations. Due to the all-to-all nature of self-attention, the computational complexity of such bidirectional video transformers is given by $\mathcal{C}_{\text{bi}}$ in Equation C1. Here we can amortize the computational complexity as the number of dot-products computed by a single self-attention Vaswani et al. (2017) operation in the Transformer network. As we know, the number of dot-products are equal to the square of the number of tokens input to the attention operation, due to the all-to-all nature of the operation. In contrast, a causal frame-by-frame transformer operating on the same latent size has complexity $\mathcal{C}_{\text{causal}}$, as derived in Equation C5. Since causal transformers achieve approximately a 2× reduction in compute compared to fully bidirectional counterparts (see eq. C6), Pyramidal-Flow Jin et al. (2025) emerges as a desirable foundation model for our system. Apart from the 2× speedup, the ability to generate videos in a streaming fashion on the fly is another characteristic of the causal attention transformer which adds to the desirability. Now going one step further, we show that the temporal pyramidal conditioning for the causal frame-by-frame generation of Pyramidal-Flow, actually gives us 32× compute saving as opposed to the 2× saving of vanilla non-pyramidal causal model (see fig. D1).

$$\textbf{Bidirectional Attention:} \quad \mathcal{C}_{\text{bi}} = (hwt)^2 \tag{C1}$$

$$\textbf{Causal Attention:} \quad \mathcal{C}_{\text{causal}} = \sum_{k=1}^{t} \underbrace{(h \cdot w)}_{\text{tokens in frame } k} \times \underbrace{(h \cdot w \cdot k)}_{\text{tokens in frames } 1..k} \tag{C2}$$

$$= \sum_{k=1}^{t} (hw)^2 \cdot k \tag{C3}$$

$$= (hw)^2 \sum_{k=1}^{t} k \tag{C4}$$

$$= (hw)^2 \cdot \frac{t(t+1)}{2} \tag{C5}$$

$$\tag{C6}$$

$$\textbf{Speedup}_{\textbf{causal}} \;=\; \frac{\mathcal{C}_{\text{bi}}}{\mathcal{C}_{\text{causal}}} \;=\; \frac{(hw)^2 t^2}{(hw)^2 \cdot \frac{t(t+1)}{2}} \;=\; \frac{2t}{t+1} \;\approx\; 2\times \text{ as } t \to \infty \tag{C7}$$

*Temporally Pyramidal Causal* **latent generation (general $S$; $t > S$).** We use a temporal pyramid with $S$ stages indexed by $i \in \{0, \ldots, S-1\}$. Stage $i$ corresponds to a spatial resolution that is downsampled by $2^i$ per dimension relative to the highest resolution. If a full-resolution frame has $M = h \cdot w$ tokens, then the number of tokens contributed by a frame at stage $i$ is

$$M_i \;=\; \frac{M}{4^i}, \qquad i = 0, 1, \ldots, S-1,$$

since each $2\times$ reduction per spatial dimension reduces the token count by a factor of $4$. We refer to stage $0$ as the highest (full-resolution) stage and stage $S-1$ as the lowest stage. For a query at frame $k$ and a history frame $j \le k$, let the temporal distance be $d := k - j$. The number of tokens contributed by this particular history frame are:

$$T(d) \;=\; \begin{cases} M, & d = 0 \quad \text{(self)}, \\[2mm] \dfrac{M}{4^{d-1}}, & 1 \le d \le S - 1, \\[3mm] \dfrac{M}{4^{S-1}}, & d \ge S. \end{cases}$$

Each query frame has $M$ query tokens, so the dot-product cost contributed by a $(k, j)$ pair is $M \cdot T(d)$. Summing over all ordered pairs $(k, j)$ with $1 \le j \le k \le t$ is equivalent to summing over distances $d$ and counting how many pairs have that distance: for a fixed $d$, there are exactly $(t - d)$ pairs $(k, j)$ with $k - j = d$.

**Total complexity (general $S$).** Let $r = \frac{1}{4}$ be the token downsampling factor, and define the finite sums

$$A(S) := \sum_{m=0}^{S-2} r^m = \frac{1 - r^{S-1}}{1 - r} = \frac{4}{3}\left(1 - 4^{-(S-1)}\right),$$

$$D(S) := \sum_{d=1}^{S-1} d\, r^{d-1} = \frac{1 - (S)r^{S-1} + (S-1)r^S}{(1 - r)^2}.$$

With $u := t - S$ (and $t > S$ so $u \ge 1$), we obtain

$$\mathcal{C}_{\text{pyr}}(t, S) = \sum_{k=1}^{t}\sum_{j=1}^{k} M \cdot T(k - j) = M^2 \left[ \underbrace{t}_{\text{self } (d=0)} + \underbrace{\sum_{d=1}^{S-1}(t - d)\, r^{d-1}}_{\text{geometric ramp } (d=1..S-1)} + \underbrace{r^{S-1}\sum_{d=S}^{t-1}(t - d)}_{\text{bulk at lowest stage } (d \ge S)} \right]$$

$$= M^2\left[ t \;+\; t\, A(S) \;-\; D(S) \;+\; \frac{u(u + 1)}{2 \cdot 4^{S-1}} \right]. \tag{C8}$$

*Asymptotically* as $t \to \infty$ (with fixed $S$),

$$\mathcal{C}_{\text{pyr}}(t, S) = \frac{M^2}{2 \cdot 4^{S-1}}\, t^2 \;+\; \mathcal{O}(t), \qquad \Longrightarrow \qquad \text{Speedup}_{\text{pyr}}(S) = \frac{\mathcal{C}_{\text{bi}}}{\mathcal{C}_{\text{pyr}}(t, S)} \;\xrightarrow[t \to \infty]{}\; 2 \cdot 4^{S-1}. \tag{C9}$$

**Specialisation to $S = 3$ (matches Fig. D1).** For $S = 3$, equation C9 becomes:

$$\textbf{Speedup}_{\textbf{temporal}} = \text{Speedup}_{\text{pyr}}(S = 3) = 2 \cdot 4^{(3-1)} = 32\times. \tag{C10}$$

The $32\times$ compute saving from the *Temporally Pyramidal Causal* attention is already a major boost, but the Pyramidal-Flow model goes further by also denoising each frame in a *spatial pyramid* (coarse-to-fine) fashion. This spatial pyramidal structure is orthogonal to the temporal pyramid and provides an additional speedup. We now derive this spatial speedup and then combine it with the temporal speedup to obtain the total compute savings over full bidirectional attention (see Fig. D1).

***Spatial pyramid setup.*** Assume that the denoising process allocates fractions $p_i$ of the total denoising steps to each stage, with $\sum_{i=0}^{S-1} p_i = 1$.

**Per-frame cost scaling.** At stage $i$, both the query and the effective K/V token counts scale by $1/4^i$ relative to full resolution. Since attention cost is bilinear in queries and keys, the per-frame cost at stage $i$ scales as

$$\text{Cost factor at stage } i \;\propto\; \frac{1}{4^i} \cdot \frac{1}{4^i} = \frac{1}{16^i}.$$

Thus, if $\mathcal{C}_{\text{temp}}^{(k)}$ denotes the per-frame cost under the *Temporally Pyramidal Causal* setup (with queries at full resolution), then the spatially adjusted per-frame cost is

$$\mathcal{C}_{\text{spatial-temp}}^{(k)} = \left( \sum_{i=0}^{S-1} \frac{p_i}{16^i} \right) \mathcal{C}_{\text{temp}}^{(k)} = \beta_S(\mathbf{p})\, \mathcal{C}_{\text{temp}}^{(k)}, \qquad \beta_S(\mathbf{p}) := \sum_{i=0}^{S-1} \frac{p_i}{16^i}.$$

**Spatial speedup.** The relative compute multiplier in the spatial dimension is $\beta_S(\mathbf{p}) < 1$, so the speedup is

$$\boxed{\text{Speedup}_{\text{spatial}}(\mathbf{p}) = \frac{1}{\beta_S(\mathbf{p})}}.$$

**Uniform allocation across stages.** If denoising steps are split uniformly across stages, $p_i = \frac{1}{S}$, then

$$\beta_S = \frac{1}{S} \sum_{i=0}^{S-1} \frac{1}{16^i} = \frac{1}{S} \cdot \frac{1 - 16^{-S}}{1 - \frac{1}{16}} = \frac{16}{15S}\left( 1 - 16^{-S} \right), \qquad \text{Speedup}_{\text{spatial}} = \frac{15S}{16(1 - 16^{-S})}.$$

For large $S$, this approaches $\beta_S \approx \frac{16}{15S}$ and $\text{Speedup}_{\text{spatial}} \approx \frac{15}{16}S$.

**Specialisation to $S = 3$.** With three spatial stages and uniform allocation $p_i = \frac{1}{3}$,

$$\beta_3 = \frac{1}{3}\left( 1 + \frac{1}{16} + \frac{1}{256} \right) = \frac{273}{768} \approx 0.3555, \qquad \textbf{Speedup}_{\textbf{spatial}} = \frac{1}{\beta_3} = \frac{768}{273} \approx 2.81\times.$$

**Combined spatio-temporal speedup.** The temporal pyramid (with $S = 3$ stages) yields an asymptotic speedup of

$$\textbf{Speedup}_{\textbf{temporal}} \approx 32\times$$

relative to full bidirectional attention. The spatial pyramid (with $S = 3$ stages and uniform allocation) yields

$$\textbf{Speedup}_{\textbf{spatial}} \approx 2.81\times$$

relative to the temporal-only baseline. Since these optimisations act on orthogonal dimensions (temporal vs spatial), the combined speedup is multiplicative:

$$\boxed{\textbf{Speedup}_{\textbf{combined}} \approx 32 \times 2.81 \approx 90\times.}$$

Thus, a *Spatio-temporally Pyramidal Causal* latent generation setup can reduce the dominant attention complexity by nearly two orders of magnitude (90×) compared to a full-resolution, fully bidirectional attention. These efficiency gains are not merely theoretical; they enable practical scaling of autoregressive video diffusion to longer sequences and higher resolutions without prohibitive compute costs. For this reason, we adopt **Pyramidal-Flow** Jin et al. (2025) as the foundation for our **Neodragon** system, leveraging its hierarchical structure to deliver both computational efficiency and strong generative performance.

For the pre-super-res output, the spatial resolution targeted in our setup is `[320×512]`. Thus the starting point for us is the Pyramidal-Flow model, which achieves a total VBench score of **80.31**. This score is obtained using the inference parameters provided by the authors in their released code, applied to the 480p low-resolution checkpoint—note that no official VBench score is reported for this checkpoint. The officially reported score for Pyramidal-Flow corresponds to its

higher-resolution 720p checkpoint, which achieves **81.72**. It is important to acknowledge that the Pyramidal-Flow model, which we adopt as our foundation, was trained with comparatively fewer GPU hours than some of the more resource-intensive open-source models such as Wan Wan et al. (2025), CogVideoX Yang et al. (2025), and LTX-Video HaCohen et al. (2024). Consequently, any inherent limitations in Pyramid-Flow's generations are inherited by our system. However, the suite of optimisations we introduce to enable efficient deployment on our platform is broadly applicable and can be extended to other models as well.

## D  PYRAMIDAL FLOW-MATCHING FOR VIDEO GENERATION

It is fascinating to observe how the training objectives for diffusion models have evolved—from early **Score Matching** approaches (Hyvärinen, 2005; Song & Ermon, 2019), which required on the order of 1,000 denoising steps for generation, to the current state-of-the-art **Flow-Matching** methods (Lipman et al., 2023; Heitz et al., 2023; Liu et al., 2023), which enable high-quality synthesis in as few as 50 steps. Along this trajectory, numerous influential works (Ho et al., 2020; Song et al., 2021a;b; Karras et al., 2022) have contributed to making diffusion training both simpler and more scalable.

**Flow-Matching.** Modern video diffusion models adopt a remarkably straightforward yet highly scalable training algorithm, capable of handling datasets with upwards of $\sim$100M videos. Starting from video latents $\boldsymbol{z} \sim p_{\text{data}}(\boldsymbol{z})$ extracted via a fixed codec-latent-VAE, we construct noisy samples as

$$\tilde{\boldsymbol{z}}_\sigma = (1 - \sigma)\boldsymbol{z} + \sigma\epsilon,$$

where $\sigma \sim \mathbb{U}(0, 1)$ denotes the noise level (0 = clean, 1 = fully noisy) and $\epsilon \sim \mathcal{N}(0, \mathbb{I})$ is Gaussian noise. This defines a continuous probability flow over $\tilde{\boldsymbol{z}}_\sigma$, whose instantaneous velocity is given by

$$v(\tilde{\boldsymbol{z}}_\sigma) = \frac{d\tilde{\boldsymbol{z}}_\sigma}{d\sigma} = \epsilon - \boldsymbol{z}.$$

Interestingly, this velocity does not depend on $\sigma$ due to the linearity of the flow, in contrast to earlier formulations such as DDPM Ho et al. (2020). The MMDiT denoiser $\mathcal{D}$ is trained to predict these velocities, conditioned on the noise level $\sigma$, via the Flow Matching objective:

$$\mathcal{L}_{\text{FM}} = \mathbb{E}_{\sigma, \boldsymbol{z}, \epsilon}\big[\|\mathcal{D}(\tilde{\boldsymbol{z}}_\sigma, \sigma) - v(\tilde{\boldsymbol{z}}_\sigma)\|_2^2\big].$$

Once trained, generation reduces to solving the ODE

$$\boldsymbol{z} = \epsilon - \int_{\sigma=1}^{0} \mathcal{D}(\tilde{\boldsymbol{z}}_\sigma, \sigma)\, d\sigma,$$

typically using a first-order solver such as discrete Euler with 50 steps, though higher-order solvers are also applicable. In practice, $\mathcal{D}$ is further conditioned on text prompt embeddings $\hat{t}$ and *CLIP* embeddings $\hat{c}$[1].

**Pyramidal Flow-Matching** (Jin et al., 2025) decomposes the probability flow into $S$ stages, where the $i^{\text{th}}$ stage ($i \in \{0, \ldots, S-1\}$) operates at a resolution that is $2^i$-times smaller than the original. Let $\text{Down}(\cdot, s)$ and $\text{Up}(\cdot, s)$ denote downsampling and upsampling by a factor $s$, respectively. Each stage is parameterised by a pair of noise levels $(\sigma_{\text{start}}^i, \sigma_{\text{end}}^i)$ with $1 > \sigma_{\text{start}}^i > \sigma_{\text{end}}^i > 0$, and operates on latents at resolution $\text{Down}(\boldsymbol{z}, 2^i)$. The start and end distributions for stage $i$ are defined as

$$\tilde{\boldsymbol{z}}_{\sigma_{\text{start}}^i} := (1 - \sigma_{\text{start}}^i)\, \text{Up}\big(\text{Down}(\boldsymbol{z}, 2^{i+1}), 2\big) + \sigma_{\text{start}}^i\, \epsilon, \quad\quad \text{(D1)}$$

$$\tilde{\boldsymbol{z}}_{\sigma_{\text{end}}^i} := (1 - \sigma_{\text{end}}^i)\, \text{Down}(\boldsymbol{z}, 2^i) + \sigma_{\text{end}}^i\, \epsilon, \quad\quad \text{(D2)}$$

where $\epsilon \sim \mathcal{N}(0, \mathbb{I})$. By the universality of the Flow Matching objective, the stage-wise loss $\mathcal{L}_{\text{FM}}^i$ is well defined to learn the flow between the above start and end distributions at stage $i$. A different local noise-level $\sigma_{\text{local}}^i \sim \mathbb{U}(0, 1)$ is used to learn the Flow-Matching model at the $i^{\text{th}}$ stage, and the global noise-level $\sigma^i$ relates to the local noise-level $\sigma_{\text{local}}^i$ as $\sigma^i = \sigma_{\text{start}}^i$. Thus, the overall Pyramidal Flow Matching objective is an aggregate over the stagewise objectives:

$$\mathcal{L}_{\text{pyr-FM}} := \sum_{i=0}^{S-1} \mathcal{L}_{\text{FM}}^i.$$

---

[1]For clarity, we omitted explicit timestep conditioning in earlier sections.

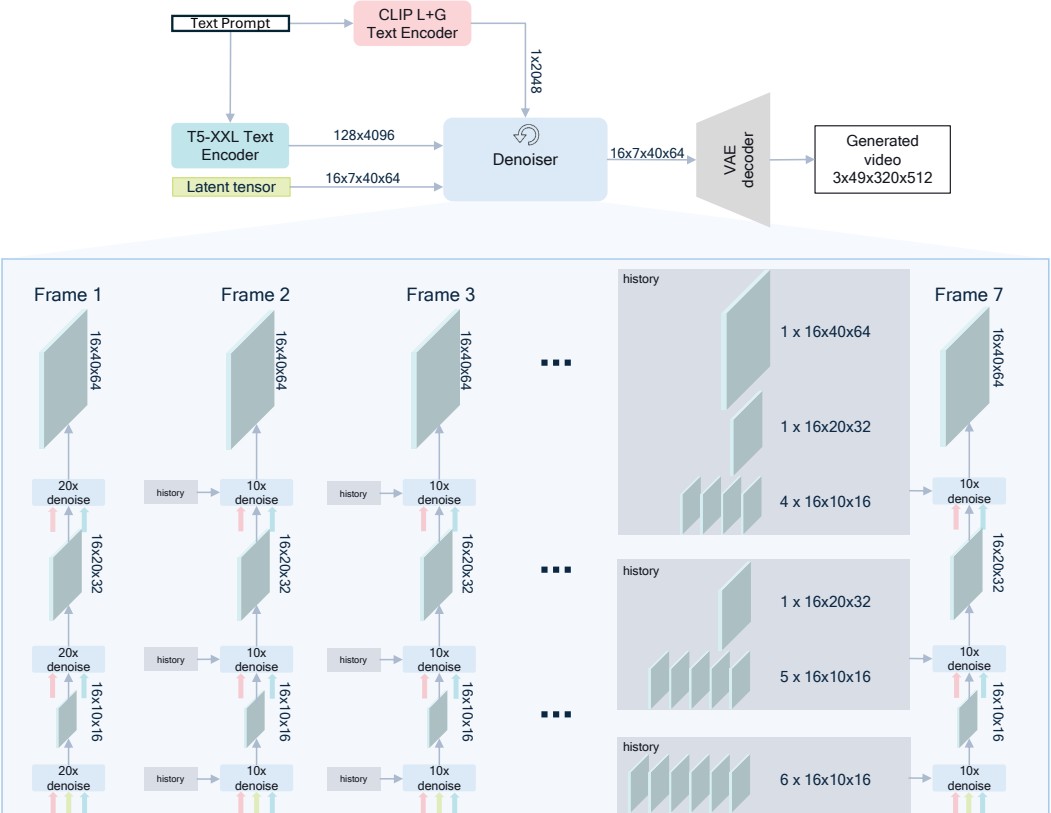

**Figure D1: Overview of the Pyramidal Autoregressive Video Diffusion Pipeline**. The pyramidal autoregressive video diffusion scheme (Jin et al., 2025) differs from the conventional latent-diffusion in how the the latent-video frames are generated (iteratively denoised). The latent frames are autoregressively generated one-by-one by denoising the curent frame while conditioning on the past history. A spatio-temporal pyramid is applied in the denoising process as: firstly the denoising of the current frame starts from a lower resolution and proceeds to reach the highest native latent-resolution; and secondly, each denoising step is conditioned on past history, where the frames from the further past are spatially downsampled.

Intuitively, the Flow-Matching model at $i^{\text{th}}$ stage flows from a noisy and pixelated version of the latent video to less noisy and less pixelated version. Note that the model is not only learning the denoising objective, but also the super-resolution objective when matching the ground truth instantaneous flow. The most ingenious contribution from Pyramidal-Flow is that the noise-levels of the different stages are not disjoint, but overlapping, which are obtained as:

$$\sigma_{\text{end}}^i = i.\frac{1}{S} \tag{D3}$$

$$\sigma_{\text{start}}^i = \frac{2\sigma_{\text{end}}^{i+1}}{1 + \sigma_{\text{end}}^{i+1}} \tag{D4}$$

And thus, after training, the same MMDiT denoiser network $\mathcal{D}$ can be used to generate the samples by flowing through all the stages and jumping resolution across stages using the following equations:

$$\tilde{z}_{\sigma_{\text{start}}^i} = \frac{2 - \sigma_{\text{start}}^i}{2}\ \text{Up}\big(\tilde{z}_{\sigma_{\text{end}}^{i+1}},\, 2\big) + \frac{\sigma_{\text{start}}^i \sqrt{3}}{2}\,\epsilon \tag{D5}$$

such that, $\epsilon' \in \mathcal{N}(0, \Sigma')$ and $\Sigma'_{\text{block}} = \begin{bmatrix} 1 & \gamma & \gamma & \gamma \\ \gamma & 1 & \gamma & \gamma \\ \gamma & \gamma & 1 & \gamma \\ \gamma & \gamma & \gamma & 1 \end{bmatrix}$ \tag{D6}

In practice (code), the value for gamma has been set to $\gamma = -1/3$ for minimising the noise added to the upsampled pixels in the neighbourhood of $2 \times 2$ after nearest neighbour upsampling in order

to continue the probability flow trajectory. We note that this specific solution is obtained for nearest neighbour upsampling where the pixel values are replicated as the reoslution is increased. Although deriving this relation for other interpolation schemes such as bilinear and bicubic could be an interesting academic exercise, we stick to using nearest neighbour due to the demonstrated high-quality empirical results by Pyramidal-Flow (Jin et al., 2025)–Ockham's Razor in action.

# E    MORE DETAILS ABOUT TEXT-ENCODER DISTILLATION

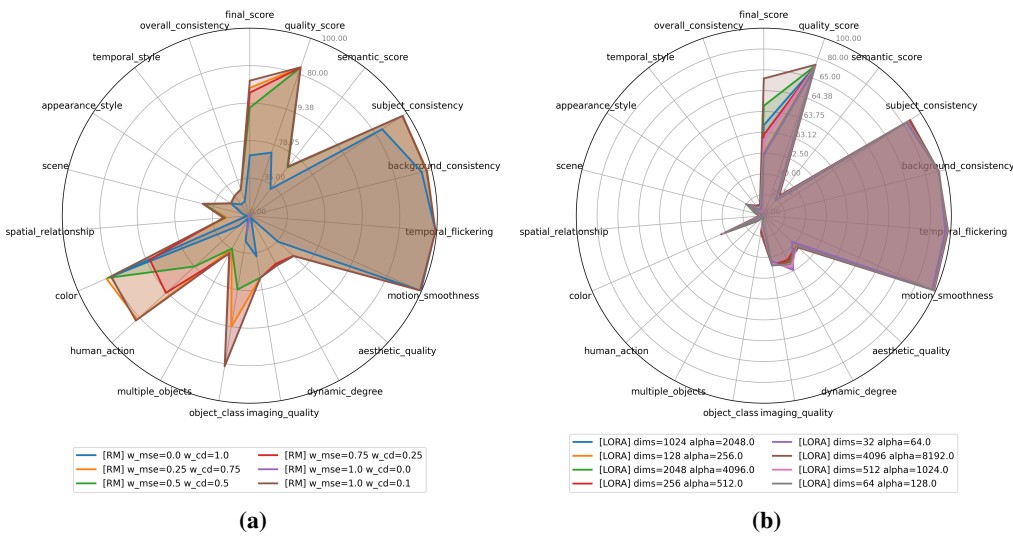

**(a)**                                    **(b)**

**Figure E1: Ablations for Text-Encoder Distillation.** We ablate the loss weights $w_{\mathrm{mse}}$ and $w_{\mathrm{cd}}$ for the **[RM]** mode in **(a)**; and ablate the two controllable hyperparameters of the LoRA layers, namely dimensions (`dims`) and the scale (`alpha`) of **[LORA]** mode in **(b)**.

Our proposed setup was trained using the Adam optimiser with a relatively higher learning rate of $3e - 3$, but decayed via a cosine schedule to $3e - 5$. Training was conducted over 24,000 iterations on four 80GB H100 GPUs, with a batch size of 512 per GPU, resulting in a total batch size of 2048. The complete training process took approximately 16 hours. As our default value, we set $w_{\mathrm{mse}} = 1.0$ and $w_{\mathrm{cd}} = 0.1$, but provide the ablation over these for the 4LMLP **[RM]** in Figure E1a. This setup enabled efficient convergence of the distilled encoder using only text data, without requiring any image or video supervision.

In Figure E1, we present a couple of ablations of our proposed Distillation Framework. Chosen for its superior empirical performance, a natural question regarding **[RM]** that arises, is how the two distinct loss functions in Equation 1 contribute to the overall optimisation landscape. Figure E1a illustrates various combinations of weights applied to the sum of these losses. When either $w_{\mathrm{mse}}$ or $w_{\mathrm{cd}}$ is set to zero, the corresponding loss is effectively disabled. We note that disabling the cosine distance results in a divergence, as denoted by the purple curve in Figure E1a (observe the centre of the radar-plot). This underscores that the Cosine Distance loss is essential for stabilising the training. Lastly, as can be observed, the best performance is obtained when we use $w_{\mathrm{mse}} = 1.0$ and $w_{\mathrm{cd}} = 0.1$.

Next, following the analysis of loss weighting, we investigate how the architecture of the $CA$ influences the adaptability of the distilled text encoder $DT5$. A comprehensive exploration of the architectural design space is beyond the scope of this work, so we ablate the number of LoRA dimensions and the LoRA scale $\alpha$ in Figure E1b for the **[LORA]** mode. Our findings indicate that even minimal visual quality in this mode necessitates a High-Rank Approximation rather than a Low-Rank one. This observation further motivates the need for full fine-tuning of the $DT5$ model's weights, which we address in the **[TDT5]** mode.

In comparison, the Extend-Mode configuration—where the new $CA$ augments rather than replaces the original $CE$ (ContextEmbedder)—incurs a slightly higher parameter overhead due to the dual-module setup. Interestingly, this configuration yields a marginally lower VBench score than

Replace-Mode, which we attribute to the rigidity of the frozen $CE$ embedding space, potentially limiting the adaptability of the extended context representation.

For the **[LORA]** variant, we observe that achieving even minimum viable video generation quality requires a significantly increased number of low-rank dimensions and a high scalar $\alpha$ (see fig. E1b). Even with these adjustments, the resulting VBench score of 64.47, while enabling some visual fidelity, falls short of the performance achieved by other configurations. This motivated further experimentation with a trainable $DT5$ encoder (**[TDT5]**). Notably, this setup achieves a strong VBench score of **79.20** with only half the parameter count of the Replace-Mode configuration. However, we ultimately select **[RM]** as our final deployment choice, prioritising even marginal gains in VBench score to maximise generation quality, despite the higher parameter count relative to the **[TDT5]** variant. Figure 3a presents qualitative examples that align with our experimental findings. Notably, the **[LORA]** mode frequently overlooks key semantic cues from the text prompts—for instance, generating a teddy bear instead of a panda, or failing to adhere to the black-and-white constraint specified in the prompt. In contrast, the remaining three modes perform comparably, with only occasional semantic mismatches.

## F  MORE DETAILS ABOUT ASYMMETRIC DECODER DISTILLATION

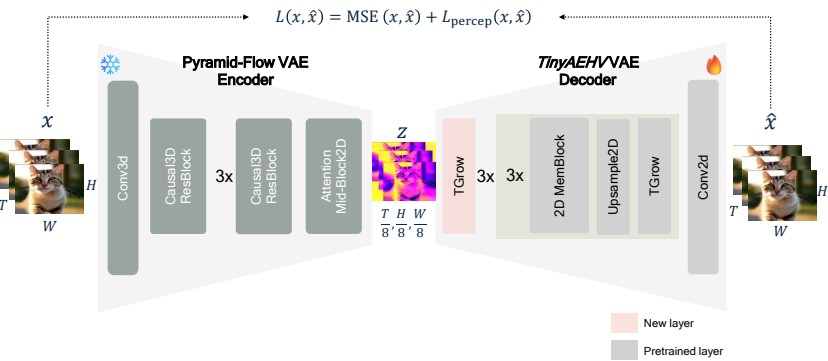

**Figure F1: Overview of the proposed Asymmetric Decoder Distillation framework.** A new asymmetric decoder from a different pretrained latent-video-diffusion model is distilled into our pipeline by: firstly modifying the decoder architecture to match the fixed [8×8×8] compressed latent-space of our model; and secondly by finetuning this asymmetric VAE with video data using MSE and LPIPS (Zhang et al., 2018) losses. The encoder is kept frozen so that the generative latent-space of the video diffusion backbone is undisturbed. We note that the *TinyAEHV* (Boer Bohan, 2025) decoder is visualised here, but the framework works with other models as well.

As shown in Figure F1, we apply minimal modifications to integrate different asymmetric decoders into our pipeline. For the TinyAEHV decoder (Boer Bohan, 2025), we modify the first `TGrow` (temporal upsampler) layer to perform 2× temporal upsampling instead of its default 1× (no) upsampling, reinitialising the parameters of this block with random weights. This single change suffices to match our latent compression factor. For the Cosmos decoder (Kwon & Buchman, 2019), we use the Continuous Tokens variant with 8×8×8 compression, requiring no architectural changes. For LTXVideo (HaCohen et al., 2024), we remove the decoder's unpatchification layer and update the `conv_out` layer with new weights. Additionally, to accommodate our 16-dimensional latents, we replace the `conv_in` layer. Finally, for the Wan decoder (Wan et al., 2025), similar to TinyAEHV, we modify the first upsampling block to perform 2× spatio-temporal upsampling instead of the default 2× spatial-only upsampling. These minimal adjustments enable us to distill diverse asymmetric decoders into out generation pipeline.

The setup was trained using the AdamW optimiser with a fixed learning rate of 1e-4. Compared to the earlier Text-Encoder Distillation experiments (ref subsec. 3.1), these runs were significantly more GPU-intensive. Training was performed for 200,000 iterations on eight 80 GB H100 GPUs, with per-GPU batch sizes ranging from 2 to 6 (depending on decoder size), resulting in an effective batch size of 16–48. We used the default patch size of [33×256×256] sampled from a corpus of approximately ∼350K videos. The full training process took about ∼120-140 hours. For the

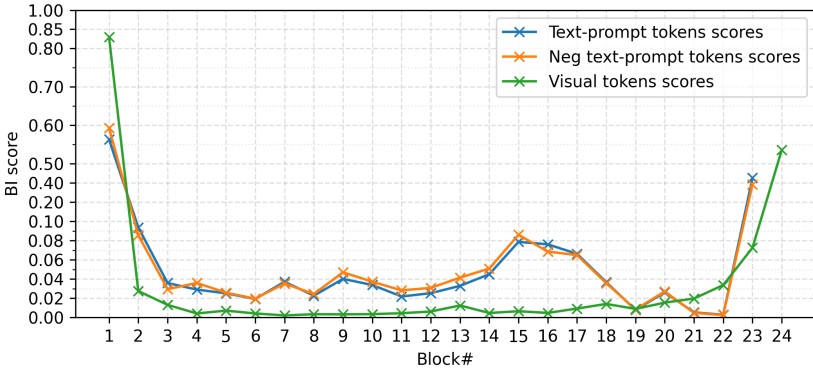

**Figure G1: Block Importance Scores v/s Block-ids.** Block Importance Scores for the 24 MMDiT blocks in the denoiser backbone, calculated using equation G5. Textual scores are computed for 23 blocks, excluding the final block where text tokens are ignored. The plot visualises token-level importance scores across two CFG forward passes: blue for descriptive text-prompts and orange for negative text-prompts. Visual token scores (green) are shown only once, as they remain identical across both passes.

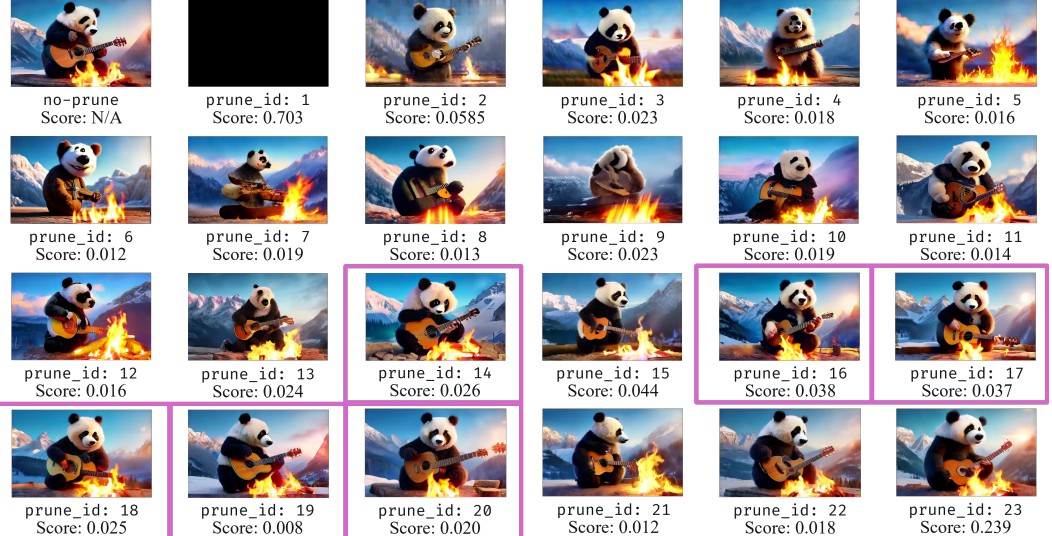

**Figure G2: Visual guidance for Block Pruning.** We visualise randomly selected frames from the generated [49×320×512] videos, across 24 different models in which the prune_id$^{\text{th}}$ MMDiT block is dropped from the model. We choose the 6 blocks highlighted with the boxes for pruning, giving us 25% model size reduction.

reconstruction objective, we followed Pyramidal-Flow's default weighting: $10.0$ for the MSE loss and $1.0$ for the LPIPS loss (Zhang et al., 2018).

## G  MORE DETAILS ABOUT MMDiT BLOCK PRUNING

**Analysing Block Importance Scores.** The MMDiT denoiser, denoted as $\mathcal{D}$, can be expressed as a composition of $N$ blocks operating on tokens concatenated from three sources: visual latent tokens $z$ (diffused with noise), textual tokens $\hat{t}$ derived from the prompt (ref eq. 2), and the clip embedding token $\hat{c}$. In our base model Pyramidal-Flow Jin et al. (2025), $N = 24$.

$$\mathcal{D}(z, \hat{t}, \hat{c}) = \mathcal{D}_N \circ \mathcal{D}_{N-1} \circ \mathcal{D}_k... \circ \mathcal{D}_1(z, \hat{t}, \hat{c}) \tag{G1}$$

$$\text{where,}\ \hat{c} := CLIP(\text{prompt}) \tag{G2}$$

Due to the multi-modal nature of the MMDiT architecture, we obtain separate block importance scores for the visual and the textual tokens, represented by $BI_k^v$ and $BI_k^v$ respectively, for the $k^{\text{th}}$

**Table G1: MMDiT Block-Pruning Stage-2 Ablations**. We ablate the choices over: the three losses, namely Feature Loss (token-matching), Teacher FM Loss (flow-matching using Teacher's predicted flow), and Data FM Loss (flow-matching using ground truth data flow); and which Block-Mapping to use when Feature Loss is active using **VBench** scores (↑).

| Feature Loss | Teacher FM Loss | Data FM Loss | Block Mapping | VBench Score Tot.(↑) | Qual.(↑) | Sem.(↑) |
|---|---|---|---|---|---|---|
| | | ✓ | N/A | 78.39 | 81.58 | 65.63 |
| | ✓ | | N/A | 80.00 | 83.52 | 65.89 |
| ✓ | | | Next-Block | 79.86 | 82.92 | 67.61 |
| ✓ | ✓ | | Next-Block | 79.93 | 82.90 | 68.02 |
| | ✓ | ✓ | N/A | 80.04 | 83.52 | 66.11 |
| ✓ | | ✓ | Next-Block | 80.35 | 83.82 | 66.48 |
| ✓ | ✓ | ✓ | Next-Block | 80.11 | 83.44 | 66.79 |
| ✓ | ✓ | | Simple | 80.10 | 83.31 | 67.28 |
| ✓ | ✓ | ✓ | Simple | 80.21 | 83.54 | 66.90 |

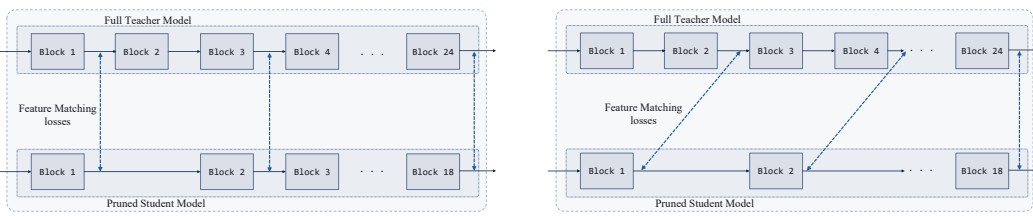

**(a)** Simple Block Mapping  **(b)** Next-Block Mapping

**Figure G3: Stage-2 Simple Block Mapping v/s Next-Block Mapping.** For minimising the token-matching losses between the Full Teacher model and the Pruned Student model, **(a)** Simple Block Mapping maps the output of each of the present block in the student model to the corresponding one in the Teacher model; whereas, **(b)** Next-Block Mapping maps the output of each present block in the Student model to the input of the next-available block in the Teacher model (except for the final block which always matches final output).

block.

$$\boldsymbol{z}_{k+1}, \hat{t}_{k+1} = \mathcal{D}_k(\boldsymbol{z}_k, \hat{t}_k, \hat{c}) \tag{G3}$$

$$BI_k^v := 1 - \mathbb{E}\left[\frac{z_k \cdot z_{k+1}}{\|z_k\|_2 \|z_{k+1}\|_2}\right] \text{ and } BI_k^t := 1 - \mathbb{E}\left[\frac{\hat{t}_k \cdot \hat{t}_{k+1}}{\|\hat{t}_k\|_2 \|\hat{t}_{k+1}\|_2}\right] \tag{G4}$$

$$BI_k := \left(BI_k^v, BI_k^t\right) \tag{G5}$$

As shown in Figure G1, we compute block-importance scores for each $k^{\text{th}}$ block in the MMDiT (see Eq. G5), defined as the Cosine Distance between the block $\mathcal{D}_k$'s input and output tokens. To estimate these scores, we use a small but diverse calibration set of 100 text prompts, generating five sample videos for each. During this process, we probe the internal token representations of the MMDiT $\mathcal{D}$ at every denoising step for both CFG (Classifier-Free Guidance Ho & Salimans (2021)) passes—one with descriptive prompts and one with negative prompts. Consistent with observations for SANA-1.5 Xie et al. (2025), we find that the initial and final blocks are more influential, while intermediate blocks contribute less, as they induce minimal residual changes to the tokens. Interestingly, due to the model's multi-modal nature, the visual and textual importance of a block are not correlated. Therefore, as illustrated in Figure G2, we also assess the impact of removing each block on the final generation quality. Based on both the importance scores and visual impact, we select six highlighted blocks for pruning, while also experimenting with smaller and slightly larger sets to explore the trade-off between quality and model size.

**Stage-1 Finetuning.**  This setup was trained using the Adam optimiser with a fixed learning rate of 3e-5 on four 80GB NVIDIA H100 GPUs, with a per-GPU batch size of 4, resulting in an effective batch size of 16. Since training was limited to 300 iterations, the process took only about 1–2 hours. Although we experimented with longer training (up to 3K iterations), we observed no significant

Original   Textual scores   Visual scores   Average scores   Visual guidance

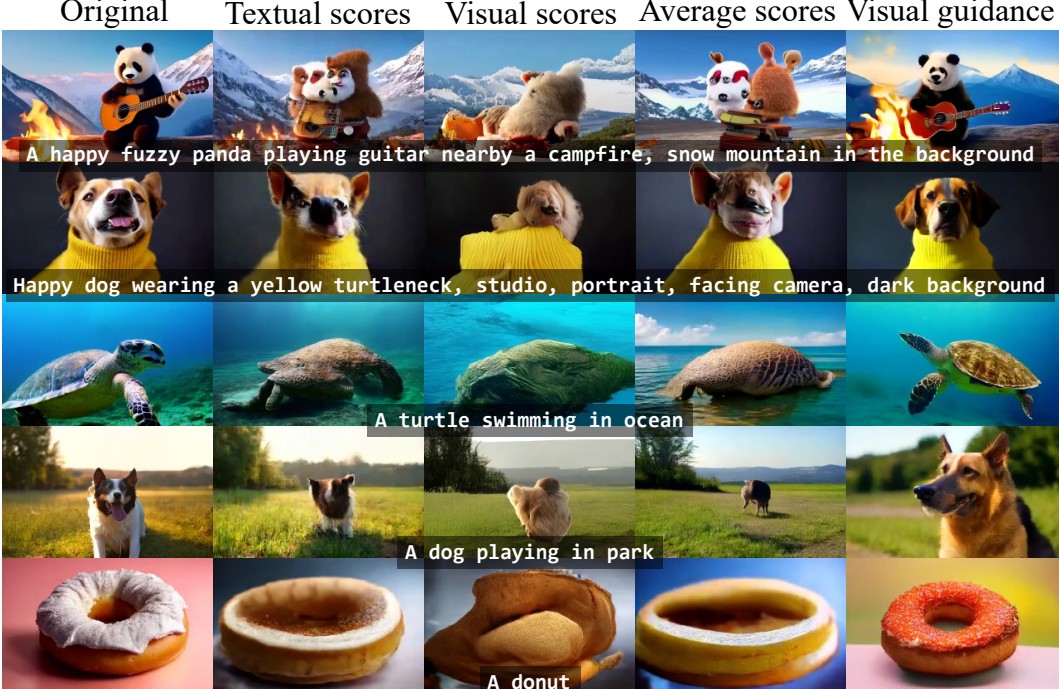

A happy fuzzy panda playing guitar nearby a campfire, snow mountain in the background

Happy dog wearing a yellow turtleneck, studio, portrait, facing camera, dark background

A turtle swimming in ocean

A dog playing in park

A donut

**Figure G4: Different BI-scores based block-pruning compared to visual guidance.** We visualise randomly selected frames from the generated [49×320×512] videos corresponding to the adjacent text prompts, across different 18-blocks pruned versions of the original model. Using only the textual-scores, or visual-scores or even average of the two (see fig. G1) directly results in a lot of semantic distortion in the generated samples as first stage finetuning. Whereas, upon choosing the blocks to prune based on the average scores as well as the visual impact (see fig. G2), causes minimal semantic distortion.

performance gains. Remarkably, even with such minimal fine-tuning, we were able to recover most of the lost performance, underscoring the effectiveness of our block selection strategy based on importance scores and visual inspection.

Also, as shown in Figure G4, we apply Stage-1 Finetuning to the 18-blocks model obtained by removing the 6 blocks using either just the textual scores, the visual scores and taking an average of them. We can clearly see from the figure how much difference the use of visual impact makes on the the the Stage-1 finetuned model.

**Stage-2 Finetuning.** After establishing a lower bound on the performance achievable by the pruned models in Stage-1, we proceed with Stage-2 finetuning of the pruned model. In this stage, we incorporate feature-matching losses between the Full Teacher model and the pruned Student model. Specifically, we apply an MSE loss on visual tokens, a cosine distance loss on textual features, and Flow-Matching losses Lipman et al. (2023) using both the Teacher model's outputs and ground-truth flow from the data as supervision. Table 2 reports the scores obtained after Stage-2 finetuning (second section). Notably, the 25% pruned model with 18 blocks achieves a VBench score of **80.21**, only 0.1% lower than the base 24-block model (**80.31**), enabling near-lossless compression of the MMDiT denoiser. Figure 4, shows qualitative examples of the kind of generations that can be obtained by the differently sized pruned models after the Stage-1 and Stage-2 finetuning.

Table G1 presents an ablation study on the design choices for loss functions used during Stage-2 finetuning. We also compare two block-mapping strategies: Simple Block mapping and Next-Block mapping, which determine how each block in the pruned Student model is paired with a block in the Full Teacher model for applying feature-matching losses. Figure G3 illustrates the difference between these two schemes. All experiments are conducted on the 18-block model. While all reasonable configurations perform well in principle, some yield slightly better results in practice. Interestingly, the model trained with Next-Block Mapping and only data-based Flow-Matching loss achieves the highest VBench score of **80.35** (even surpassing the base model). However, this config-

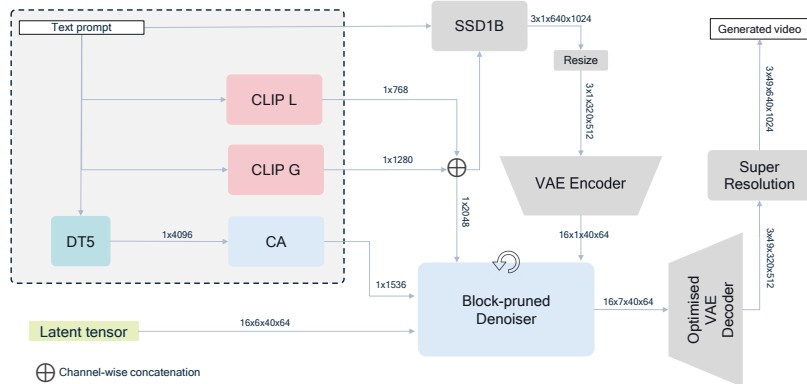

**Figure H1: Neodragon E2E full pipeline**. **Neodragon** integrates all four of our proposed optimisations namely, Distilled small Text-Encoder 3.1, Asymmetric VAE Decoder 3.2, Block-Pruned MMDiT 3.3, and Step-Distilled scheduler 3.4 (not shown here). For boosting the visual fidelity of the generations, we also include SSD1B (Gupta et al., 2024) for generating the first image, and QuickSRNet (Berger et al., 2023) for 2x super-resolution.

uration introduces artifacts in some generations and occasionally produces black videos for certain prompts. Therefore, we adopt the model trained with all losses and Simple Block Mapping as the final version for deployment in the Neodragon pipeline (see Fig. H1). As reported in Table 2, starting from the full 24-block MMDiT with a Qualcomm Hexagon NPUlatency of **1.15s**, we reduce the latency to **0.92s** with minimal impact on VBench performance.

## H  MORE DETAILS ABOUT STEP DISTILLATION

To further accelerate generation, we explore diffusion step-distillation techniques in this section, which aim to reduce the NFE requirement of the Denoising Scheduler. To this end, we begin by detailing the training objective of *Pyramidal* Flow-/Matching (Jin et al., 2025), followed by an explanation of how we adapt four different Flow-Matching step-distillation techniques from the literature to this pyramidal setting. The selected techniques include DMD (Yin et al., 2024), Direct Progressive Distillation (Salimans & Ho, 2022), and Adversarial Distillation (Sauer et al., 2024) as discrete step-distillation methods, and the recent Mean-Flows (Geng et al., 2025) as a continuous consistency-based approach. While Mean-Flows was originally proposed for training models from scratch, we adapt it to operate in a distillation setting by applying it to an already trained model.

**Pyramidal Mean-Flows** Mean-FLows Geng et al. (2025) proposed two changes to the learning algorithm of Flow-Matching models. First, they propose to model not the instantaneous velocity field $v(\tilde{z}_\sigma)$ of the underlying probability-flow ODE, but instead the *Mean* velocity field $v_{\text{mean}}(\tilde{z}_\sigma, \beta, \sigma)$ which denotes the average velocity of the trajectory going from $\beta$ to $\sigma$ (such that, $\sigma > \beta$). A direct implication of which is that the denoiser network now needs to also condition on $\beta$ apart from $\sigma$, i.e. the mean-predicted velocity is now $\mathcal{D}(\tilde{z}_\sigma, \beta, \sigma)$. Through a very interesting derivation, Mean-Flows shows that the learning objective for such a Mean-Flows model is given by:

$$\mathcal{L}_{\text{MF}} = \mathbb{E}_{\sigma,\beta,\boldsymbol{z},\epsilon}\left[\|\mathcal{D}(\tilde{z}_\sigma, \beta, \sigma) - v_{\text{mean}}(\tilde{z}_\sigma, \beta, \sigma)\|_2^2\right].$$

Where, the target ground-truth $v_{\text{mean}}$ is computed as:

$$v_{\text{mean}}(\tilde{z}_\sigma, \beta, \sigma) = v(\tilde{z}_\sigma, \sigma) - (\sigma - \beta)(v(\tilde{z}_\sigma, \sigma)\partial_{\boldsymbol{z}}\mathcal{D} + \partial_\sigma\mathcal{D})$$

Where the latter term is computed as a `JVP` in code. In Our Pyramidal Mean-Flows version, we extend the Pyramidal Flow Matching loss for each of the $i^{\text{th}}$ stage such that the $v_{\text{mean}}$ is computed as,

$$v_{\text{mean-pyr}} := v(\tilde{z}_\sigma, \sigma) - (\sigma - \beta)(v(\tilde{z}_\sigma, \sigma)\partial_{\boldsymbol{z}}\mathcal{D} + (\sigma_{\text{start}}^i - \sigma_{\text{end}}^i)\partial_\sigma\mathcal{D})$$

Note the scaling of the last term, which accounts for the scaled version of the instantaneous velocities which are learned by the Pyramidal-Flow Matching model. Thus, with this correction to the $v_{\text{mean-pyr}}$, we can define the squared L2 loss per stage to obtain $\mathcal{L}_{\text{MF-pyr}}^i$, and then giving us the aggregate loss

function $\mathcal{L}_{\text{MF-pyr}}$. In practice of course since we are finetuning the MMDiT $\mathcal{D}$ from a pretrained Flow-Matching model, we first only finetune it with the second starting point conditioning $\beta$ while only training for Flow-Matching objective, i.e. setting $\beta = \sigma$, and later training the Mean-Flows objective. Also as a key detail, the training needs to have only 25% of the batch-samples that are Mean-Flows (i.e. $\beta \neq \sigma$), while the rest are still Flow-Matching samples so that the training stabilises. From our experiments, we found that fine-tuning the model in such a way very soon leads to a loss that is overpowered by Flow-Matching rather than Mean-Flows, while increasing more Mean-Flows batch samples leads to unstable training. Further exploring the cause of this instability constitues an interesting direction for future work.

**Pyramidal Progressive Distillation** We apply the Progressive Distillation Salimans & Ho (2022) to Pyramidal Flow-Matching. Firstly, we found that stage-wise $2\times$ distillation is not necessary for a Pyramidal Flow-Matching model, which already can do the generation with 20 steps for a single resolution, thereby doing denoising of a single frame in 60 steps (given 3 pyramidal resolutions). Thus, for per stage we setup two networks: a student network $\mathcal{D}_\theta$ (to be distilled) and the teacher network $\mathcal{D}$ for supervision. Then, we obtain synthetic generated videos for the $\sim 350K$ prompts that we had curated to form our dataset so that we never leave the support of the probability distribution that is learned by the teacher network $\mathcal{D}$.

We quantise the steps of the Teacher model to 16 uniformly sampled $\sigma^i \in \{\sigma^i_{\text{start}}, ...14 \text{ steps}..., \sigma^i_{\text{end}}\}$ values (Euler solver) for the $i^{\text{th}}$ stage. The Student then either learns a single step (for 1-1-1 configuration) or 4 steps (for 4-4-4 configuration) uniformly subsampled from the set of the teacher steps. We detail the loss objective for the 4-4-4 configuration below, but note that the 1-1-1 follows directly from it, or any other distillation configuration as long as the number of teacher steps are are perfectly divisible by the number of student steps that are to be distilled.

Given a sampled $\sigma^{is}_{\text{teach}}$, we take the next $j$ steps (in this case 4 steps) to fix the last sigma from the teacher inference as $\sigma^{is+j}_{\text{teach}}$. The ground truth end-point from the teacher inference trajectory given the starting point $\tilde{z}_{\sigma^{is}_{\text{teach}}}$ is computed by running no_grad mode Euler inference of the Teacher:

$$\tilde{z}_{\sigma^{is+j}_{\text{teach}}} = \tilde{z}_{\sigma^{is}_{\text{teach}}} + \sum_s^{s+j} (\sigma^{is+1} - \sigma^{is}) \mathcal{D}(\tilde{z}_{\sigma^{is}_{\text{teach}}})$$

Then, the student model's prediction to match the $\tilde{z}_{\sigma^{is+j}_{\text{teach}}}$ in one step is computed again as an Euler step:

$$\tilde{z}_{\sigma^{is+j}_{\text{stud}}} = \tilde{z}_{\sigma^{is}_{\text{stud}}} + (\sigma^{is+j} - \sigma^{is}) \mathcal{D}_\theta(\tilde{z}_{\sigma^{is}_{\text{stud}}})$$

Thus, having computed the noisy simulations from the Teacher as well as the student's one-step noisy predictions, the loss function for this distillation is defined as

$$\mathcal{L}_{\text{pyr-prog}} := \mathbb{E}_{\sigma, z, \epsilon} \left[ \| \tilde{z}_{\sigma^{is+j}_{\text{stud}}} - \tilde{z}_{\sigma^{is+j}_{\text{teach}}} \|_2^2 \right]$$

Intuitively, we basically run the teacher model $\mathcal{D}$ for four Euler steps to get the teacher's prediction, without gradients and then train the student model to match this output. Once distilled, the student model can then be used to run the inference in the distilled number of steps. While in all this, as explained mathematically, we ensure that the velocities are still being scaled per resolution (stage) correctly and that we use the local noise-levels in the teacher and student simulations.

**Pyramidal Adversarial Step Distillation** Finally in the Pyramidal Adversarial Step Distillation approach, we follow the same setup as the progressive one, but also add a patchwise GAN loss on top of the squared L2 loss of the Progressive distillation approach.

$$\mathcal{L}_{\text{pyr-adv}} := w_{\text{recon}} \mathcal{L}_{\text{pyr-prog}} + w_{\text{adv}} \mathcal{L}_{\text{GAN}}(\tilde{z}_{\sigma^{is+j}_{\text{stud}}}, \tilde{z}_{\sigma^{is+j}_{\text{teach}}})$$

We use the Hinge-GAN loss for the $\mathcal{L}_{\text{GAN}}$ which speeds up the distillation process by not only focussing on the pixelwise distance, but also matching the distributions of the noisy tokens adversarially.

As is standard practice with Adversarial step distillation, we use the Features extracted from the Teacher network $\mathcal{D}$ passed to a 4layer MLP as the Discriminator architecture for the GAN loss. The teacher network always remains frozen and only the MLP is trainable. We empirically found the values of $w_{\text{recon}} = 10.0$ and $w_{\text{GAN}} = 1.0$ to converge well.

| Prompt Dimension | Human Preference % | | |
|---|---|---|---|
| | Neodragon | No preference | PyramidalFlow |
| Appearance Style | 26.9 | 38.5 | 34.6 |
| Color | 61.5 | 38.5 | 0.0 |
| Human Action | 3.8 | 73.1 | 23.1 |
| Multiple Objects | 73.1 | 23.1 | 3.8 |
| Object Class | 84.6 | 15.4 | 0.0 |
| Overall Consistency | 57.7 | 23.1 | 19.2 |
| Scene | 38.5 | 23.1 | 38.5 |
| Spatial Relationship | 15.4 | 76.9 | 7.7 |
| Subject Consistency | 53.8 | 34.6 | 11.5 |
| Temporal Flickering | 53.8 | 30.8 | 15.4 |
| Temporal Style | 63.5 | 9.6 | 26.9 |
| Total | 48.3 | 29.8 | 21.8 |
| | Neodragon | No preference | Hummingbird |
| Appearance Style | 69.2 | 10.3 | 20.5 |
| Color | 96.2 | 1.9 | 1.9 |
| Human Action | 80.8 | 19.2 | 0.0 |
| Multiple Objects | 71.8 | 25.6 | 2.6 |
| Object Class | 90.4 | 5.8 | 3.8 |
| Overall Consistency | 61.5 | 23.1 | 15.4 |
| Scene | 94.2 | 5.8 | 0.0 |
| Subject Consistency | 92.3 | 7.7 | 0.0 |
| Temporal Style | 76.9 | 23.1 | 0.0 |
| Total | 83.7 | 12.0 | 4.3 |

**Table I1:** Results of the method-blinded human visual preference study over 750 paired video comparisons. Rows correspond to subsets filtered by different VBench prompt dimensions.

**Table J1: Neodragon Quantization Scheme Used**. We report the quantization scheme used for each modules here.

| Module | CLIP L | CLIP G | DistilT5 | SSD1B | | VAE Enc. | VAE Dec. | MMDiT+CA | | | QuickSRNet |
|---|---|---|---|---|---|---|---|---|---|---|---|
| | | | | UNet | Dec. | | | [7×10×16] | [7×20×32] | [7×40×64] | |
| Quant Scheme | FP16 | FP16 | FP16 | W8A16, PTQ | W8A16, PTQ | W8A16, PTQ | W8A16, PTQ | W8A16, PTQ | W8A16, PTQ | W8A16, PTQ | W8A16, AdaRound |
| Calibration Data Size | NA | NA | NA | 500 | 500 | 50 | 50 | 300 | 300 | 300 | 500 |
| deploy SNR | NA | NA | NA | 33dB | 31dB | 40dB | 35dB | 29dB | 22dB | 24dB | 48dB |

# I    MORE DETAILS ABOUT END-TO-END INTEGRATION

Figure  H1 details our full end-to-end pipeline that we ship as part of the Neodragon Mobile VDM. It provides a highly modular mechanism for generating videos on device.

## USER STUDIES

Table I1 summarizes the results of a method-blinded human preference study. In this study, we presented participants with unlabeled video pairs generated by two competing methods using identical input prompts. Participants were asked to select their preferred video or indicate no significant difference as a third option. To minimize potential bias, the order of method appearances within each pair was randomized. The set of prompts for comparison where randomly selected from the VBench prompts set. The results indicate that Neodragon is more often preferred over Pyramidal-Flow, while its preference rate dramatically increases when compared to AMD-Hummingbird.

# J    MODEL PIPELINE QUANTIZATION

**Fixed point quantization.**    The quantization scheme used for deploying the various modules and calibration results are described in table  J1. We use *AIMET - AI Model Efficiency Toolkit* Siddegowda et al. (2022) to perform Post Training Quantization (PTQ). W8A16 quantization scheme is used for all but QuickSRNet. We apply AIMET *AdaRound* Nagel et al. (2020), a weight rounding technique, to QuickSRNet which ends up adding 7+dB SQNR. In this table, SQNR is calculated between original FP models vs. the deployed models

It's worth pointing out that the complexity of our pipeline has an impact on end-to-end integration SQNR as the quantization loss, despite minimal for each model, compounds quickly. In DiT pipeline, for example, we have FP embeddings coming from text encoders and first frame generated

by SSD1B pipeline. Then it goes through 3 stages of DiTs mutiplied by the number of timesteps run in each stage. After which it's decoded by VAE decoder and upsampled by QuickSRNet. This is where the step distillation mentioned in previous section comes to our assistance . Reduction to only 1 timestep on each stage doesn't just help reduce latency but also greatly mitigates the compounding quantization loss along the pipeline.

## K  DISCUSSION OF LIMITATIONS

Despite its promising results, our Neodragon has a few limitations that highlight opportunities for future work.

**First**, the overall performance of the system is bounded by the capabilities of the underlying Pyramidal-Flow baseline. While our optimisations significantly reduce latency and memory footprint, they do not fundamentally alter the generative capacity of the base model. Consequently, Neodragon inherits constraints on video resolution, temporal length, and semantic fidelity, as reflected in VBench scores. For instance, our pipeline currently supports videos of 49 frames at 640×1024 resolution, which, although competitive for mobile deployment, falls short of the high-resolution and long-duration outputs achievable by server-scale models such as Wan or CogVideoX. Extending Neodragon to support longer sequences and higher resolutions without compromising efficiency remains an open challenge.

**Second**, our MMDiT block pruning strategy introduces a practical bottleneck: the selection of blocks to prune currently requires some visual inspection although having the guidance from the BI scores. While our approach leverages block-importance scores and visual inspection to identify low-impact blocks, this process is labour-intensive and lacks scalability. Moreover, the multi-modal nature of MMDiT means that visual and textual importance scores are often uncorrelated, complicating automated decision-making. To address this, we are investigating heuristic-driven pruning strategies that incorporate perceptual metrics such as LPIPS, aiming to automate block selection while preserving semantic alignment and temporal consistency.

**Third**, the step distillation process—critical for reducing the number of denoising steps—remains an unresolved research problem. Our adaptation of pyramidal DMD achieves substantial speedups, reducing NFEs from 480 to 21 and cutting latency from 147s to 6.4s in the 1-1-1 configuration. However, this acceleration comes at the cost of visual artifacts and semantic inconsistencies, particularly in the first frame. To mitigate these issues, we currently employ a hybrid architecture that generates the initial frame using SSD-1B and applies motion synthesis via the block-pruned MMDiT. While effective, this workaround introduces architectural complexity and deviates from the ideal of a unified model. Developing a robust, single-model solution for step distillation that preserves fidelity across all frames is an active area of research and a key goal for future iterations of Neodragon.

In summary, Neodragon demonstrates that mobile video generation is feasible, but achieving high-resolution, long-duration, and artifact-free outputs in a fully unified architecture remains an open challenge. Addressing these limitations will require advances in automated pruning, principled step distillation, and scalable architectural design, paving the way for truly interactive and high-quality video synthesis on consumer hardware.

## L  USE OF LARGE LANGUAGE MODELS

We used Microsoft Copilot (a large language model) to aid in polishing the writing of this submission. The model was employed solely for improving clarity and readability; all ideas, technical content, and conclusions are our own.

