# OpenReview forum: "Neodragon: Mobile Video Generation Using Diffusion Transformer"
_ICLR.cc/2026/Conference — ICLR 2026 Poster_

### Official Review · Reviewer_oTLa · 2025-10-25

**Soundness:** 4
**Presentation:** 3
**Contribution:** 3
**Rating:** 8
**Confidence:** 4

**Summary:**

This paper proposes Neodragon, an efficient on-device video diffusion transformer designed to run entirely on a mobile NPU. The method starts from the Pyramidal-Flow framework and introduces four complementary optimizations:
1. Text-Encoder Distillation: Compresses the large T5-XXL encoder into a lightweight “DT5 + ContextAdapter” model with minimal quality loss.
2. Asymmetric Decoder Distillation: Replaces the original heavy VAE decoder with a much smaller decoder transferred from other models, keeping the encoder and DiT frozen.
3. MMDiT Block Pruning: Removes less important blocks from the diffusion transformer while maintaining fidelity through two-stage fine-tuning.
4. Step Distillation for Pyramidal-Flow: Reduces the number of denoising steps from 480 to 21 using an extended Distribution Matching Distillation (DMD) strategy.

These combined optimizations enable real-time text-to-video generation on the Qualcomm Hexagon NPU, setting a new state of the art for mobile video generation.

**Strengths:**

1. **Well-structured and practical problem analysis:** I appreciate the way the paper systematically analyzes bottlenecks in the video diffusion pipeline. It first identifies that memory (not compute) is the primary constraint, and addresses this through text encoder and decoder distillation. Once memory becomes manageable, the authors shift focus to speed via block pruning and denoising step distillation. This staged approach shows deep understanding of real deployment challenges and practical prioritization.
2. **Targeted and effective design choices.** Each optimization directly addresses a critical performance bottleneck—encoder size, latent reconstruction cost, model depth, and step count—without losing model compatibility. The work shows excellent engineering sense in solving exactly the key part at each stage.
3. **Comprehensive ablation studies.** Nearly every design component is supported by specific ablations. For example, Figure 1 (b) compares different text encoder distilltation method, Table 1 compares multiple decoder architectures under asymmetric fine-tuning; Table 2 analyzes MMDiT pruning ratios; and Table 3 reports performance across step distillation configurations. This makes the conclusions well supported and convincing.
4. **Rich visual evidence.** The paper provides abundant qualitative results, both in the main paper and in the appendix, including frame-by-frame comparisons and long video samples. These visuals clearly demonstrate that the model preserves temporal consistency and visual quality even after aggressive compression.

**Weaknesses:**

1. Although the experiment is rich and supportive, the setting of experiment is sometimes not clearly stated, For example, I do not realize what base model the paper use before a second read. Second example is that I was confused for a while about what "Neodragon E2E" and "T2V Multi-Step" are, and later find in line 470 and 471 what these mean. The author is encouraged to add a specific section to clarify the setting of experiment conducted throughout the paper and I believe this would greatly increase the clarity of the paper. Possibly this is due to limited spaces. Also, some table/figure (like table 4) is not referred in the text.
2. Lack of NPU Implementation Detail. While the paper reports on-device latency and memory numbers, it omits critical information such as hardware numbers (memory, flops), runtime environment, or hardware-specific optimizations.

**Questions:**

1. The paper perform full-stack, end-to-end optimization on pyramid-flow model. I wonder whether the similar techiques like can be applied--or have been tried--on other video diffusion models such as Wan or CogVideoX, which target consumer-level GPUs. Would the authors expect comparable performance gains or face new challenges due to architectural differences?
2. I am experienced in CUDA programming and are curious about NPU programming. Could the author compare the differences or GPU and NPU programming and provide some implementation insight?
3. The step distillation process seems to offer much greater acceleration than block pruning. Can the author provide some insights in how to combine step distillation and block pruning to achieve best speed accuracy tradeoff? Does the speed gain of block pruning worh the accuracy loss?

---

> ### Author Response · Authors · 2025-11-20
>
> Thanks for the insightful comments, they have been duly noted. Please find our detailed response as follows:
>
> **Confusing exposition:**
> Thank you for pointing this out and for the clear suggestion. We have corrected these exposition issues in the current revision.
>
> **More On-device deployment details:**
> Thanks for pointing this out. This has been a recurring suggestion from other reviewers as well, so we have added details to Section 4.2 (given one additional page) and Appendix Section J.
>
> **Transferability of optimisations to other models:** We believe that the optimisations of the Text-Encoder and the Decoder can be applied as-is to various other models with same (good) expected performance. Regarding the Block-Pruning, and Step-Distillation optimisations the milage may vary slightly when applied naively to models like WAN, but adapting the approach to those models shouldn't require a lot of effort and/or new solutions.
>
> **CUDA v/s NPU programming:**
> Thank you for your interest in NPU programming. In our work, we did not directly program the NPU at the kernel level. Instead, we leveraged Qualcomm’s AI Engine Direct SDK, which compiles the PyTorch computation graph into an optimized binary executable for the device. This compilation process automatically handles operator mapping and hardware-specific optimisations, so we did not need to implement any custom kernels.
>
> Compared to CUDA programming on GPUs, NPU programming typically involves stricter resource constraints (e.g., limited cache memory such as 8 MB) and a reduced instruction set. NPUs are specialised DSP-based architectures with extensions like HVX (Hexagon Vector Extension) for vector operations and HMX (Hexagon Matrix Extension) for matrix operations. Writing custom kernels for NPUs is possible but requires deep knowledge of the environment and is uncommon among third-party developers. In our case, all required operations were supported by the SDK, so no custom kernel development was necessary.
>
> **Block-Pruning and Step-Distillation tradeoff:**
> Thank you for raising this important question. While our block-pruning optimisation was primarily aimed at latency reduction, its main effect is reducing model size (#params), which in turn lowers I/O transfers between RAM and the NPU (see fTZz’s rebuttal for details). This reduction improves power efficiency and accelerates execution. Although the latency gains are smaller compared to step-distillation (184.2s → 147.4s), the benefits of a smaller model package and reduced memory I/O make block-pruning a desirable optimisation.
> Regarding step-distillation, we note that despite being well-studied, current solutions do not robustly address single-step or few-step diffusion-based generation. We adopted DMD for the Pyramidal Flow-Matching model, but its adversarial nature makes it challenging to stabilise and requires extensive hyperparameter tuning. Thus, step-distillation remains an open problem.
> Finally, combining block-pruning and step-distillation could lead to an MoE-style solution, where smaller expert variants of the base model specialise in different quantised noise levels. If designed carefully, the resulting MoE package could be smaller than the original model while achieving better performance through targeted step-distillation. We appreciate this question—it opens an exciting direction that we will consider for future work.

---

> > ### Comment · Reviewer_oTLa · 2025-11-21
> >
> > Thanks for the effort of the author. I will keep my score. I believe this paper makes a meaningful contribution and is definitely suitable for acceptance.

---

### Official Review · Reviewer_eXNS · 2025-11-01

**Soundness:** 3
**Presentation:** 3
**Contribution:** 3
**Rating:** 6
**Confidence:** 4

**Summary:**

This paper proposed Neogradon, a video DiT designed to run on a low-power NPU present in devices. It designed a lightweight text encoder, video decoder, pruned diffusion transformer, and step distillation method for applying Distribution Matching Distillation for the Pyramidal Flow-Matching objective. Experiments shown that Neodragon generates 49 frames of 640×1024 resolution videos within 7.6 seconds on the Qualcomm Hexagon NPU with a VBench total score of 81.61.

**Strengths:**

1. The motivation is clear and meaningful as a on-device design of video generation model are important for phones or laptops.

2. The experiments are thorough, providing a detailed exploration of various compression techniques.

3. The paper has adapted various components in the video generation model on the edge-device, which is a very systematic project.

4. The proposed model has good performance and can achieve advanced generation performance on the edge-device.

**Weaknesses:**

1. The display of some relational data in the paper is not intuitive, such as some key metrics for end-to-end deployment such as memory consumption and latency, which are only explained in text. Providing some tables or charts would be more obvious.

2. The paper should provide clearer references to some of the baseline choices made in the appendix regarding the baseline method, such as why Pyramidal Flow was chosen.

3. More efficiency should be reported, such as memory consumption and latency of other comparison methods.

4. The balance between the performance and efficiency of different proposed components on the final model should be reported.

**Questions:**

Please see above weaknesses.

---

> ### Author Response · Authors · 2025-11-20
>
> Thanks for the insightful comments, they have been duly noted. Please find our detailed response as follows:
>
> **Unclear metrics for end-to-end deployment:**
> Thank you for the suggestion. Due to space constraints in the initial submission, we reported device deployment metrics directly in the text. With the addition of an extra page, we have revised Section 4.2 to include more detailed on-device measurements.
>
>
> **Clarifying the choice of baseline:**
> The primary reason for selecting Pyramid-Flow as our baseline for on-device porting is its 90× compute savings compared to a vanilla fully attentive bidirectional DiT such as Wan 2.1 (1.5B). We have improved the mathematical derivation in Appendix Section C, and have made this reference more explicit in the revision. Apologies for the earlier oversight.
>
>
> **Additional measurements and comparisons to other methods:**
> We appreciate the emphasis on scientific rigour. Unfortunately, other on-device methods such as SnapGen-V and Wu et al. have not released their code or models, and porting comparative baselines like AMD-Hummingbird to mobile hardware is resource-intensive and costly process. Nevertheless, by providing detailed on-device measurements for Neodragon, we aim to promote openness and set a precedent for fair benchmarking of the future works.

---

### Official Review · Reviewer_fTZz · 2025-11-01

**Soundness:** 3
**Presentation:** 3
**Contribution:** 3
**Rating:** 4
**Confidence:** 3

**Summary:**

The paper proposes Neogradon, a video DiT (Diffusion Transformer) designed to run on a low-power NPU present in devices such as phones and laptop computers. It reduces the diffusion sampling cost using their novel extended version of DMD (Distribution Matching Distillation) for the Pyramidal Flow-Matching objective.

**Strengths:**

1. The paper successfully transforms a large, server-side Diffusion Transformer into a highly efficient, deployable mobile solution.
2. The paper is well-structured, featuring a data-driven abstract and logical presentation of the methodology.
3. The paper provides a finding that the Cosine Distance loss is indispensable for stabilizing the text encoder distillation process. This highlights the crucial role of preserving the directional coherence of text embeddings for the downstream attention mechanisms.

**Weaknesses:**

1. The final performance claim (VBench 81.61) is achieved by a hybrid pipeline that uses external models (SSD-IB for high-quality first-frame initialization and QuickSRNet for super-resolution) to compensate for artifacts caused by aggressive step distillation in the core model.
2. The key assumptions underpinning the core compression strategies, such as the "universality of compressed video latent space" and the concept of "similarly shallow semantic demands" for large language models, are supported primarily by empirical results rather than formal theoretical proofs.
3. The paper lacks critical quantitative data, such as comparisons of NPU peak memory usage and real-time decoding latency, to fully justify the final selection of certain components (e.g., choosing the TinyAEHV decoder despite its lower PSNR).
4. The necessity of the two-stage curriculum used for MMDiT block pruning is only proven empirically (i.e., direct Stage 2 fails), but a definitive theoretical or optimization-landscape explanation for why this two-step curriculum is mandatory is missing.
5. The block importance metric used for MMDiT pruning is insufficient on its own and requires the inclusion of subjective visual impact assessment to determine the final blocks to remove, suggesting the need for a more comprehensive, objective metric.

**Questions:**

For specific issues, please refer to the points listed in the Weaknesses.

---

> ### Author Response · Authors · 2025-11-20
>
> Thanks for the insightful comments, they have been duly noted. Please find our detailed response as follows:
>
> **Hybrid full-pipeline evaluation:**
> We clarify that the reported VBench score of 81.61 was achieved without using the Super-Resolution (SR) network, ensuring fair comparison with other methods operating at the same resolution. The SR module is applied only as an on-device post-processing step to qualitatively enhance fidelity and is therefore excluded from all evaluations and comparisons.
> Regarding the hybridisation approach using SSD-1B for first-frame generation to mitigate step-distillation issues: we acknowledge that this solution is not as robust as the optimisations introduced via text-encoder distillation, asymmetric decoder distillation, and MMDiT block pruning. Given the project timeline for this version, we adopted the hybrid solution as a practical compromise. We have now explicitly stated this as a limitation in Appendix Section K (in the current revision) and note that developing a more principled step-distillation method is part of our future work. Nevertheless, we believe our insight—that step-distillation in AR-hybrid diffusion models (commonly used for efficient video generation) primarily impacts first-frame generation—is valuable and can inspire further research in this area.
>
> **Empirical v/s theoretical evidence:**
> We agree that formal mathematical proofs provide strong guarantees. However, in this field, empirical evidence often serves as a more practical guide, as experiments frequently diverge from idealised theoretical assumptions and require nuanced implementation tricks to succeed. Specifically, for assessing the semantic demands generative tasks place on LLMs, to the best of our knowledge, no formal mathematical frameworks currently exist—likely reflecting the inherent difficulty of such proofs.
>
>
> **NPU and other deployment details:**
> The NPU is a highly power-efficient processor that evolved from the prior DPSs (Digital Signal Processors), and thus as such, unlike a GPU, it doesn’t have dedicated memory associated with it. It shares the global RAM present in the device (phone / laptop), and thus one of the challenges is to minimize the I/O transfers which are costly in terms of power consumption as well as latency. We have added new details to the revision in Section 4.2 and Appendix section J, please refer to the same for more details on the process of compiling and quantizing a torch model for Qualcomm Hexagon NPU. Regarding the choice of TinyAEHV, it was mainly driven by it’s simplicity, it’s parameter count, and it’s deployability to the NPU. We have clarified this explicitly in the current revision of the paper.
>
> **Investigating direct stage-2 finetuning failure:**
> Thank you for highlighting this intriguing phenomenon. We share your curiosity and are actively investigating it through both empirical experiments and theoretical analysis as part of our future work. Although we currently lack a concrete explanation, we believe this observation opens an interesting direction for the community to explore collaboratively.
>
> **BI scores limitation:**
> We acknowledge this limitation of the current Block-Pruning approach. Developing a more robust automated heuristic—such as leveraging LPIPS or similar feature-space metrics for identifying less-important blocks—is part of our ongoing work. We have made this limitation explicit in the revision Appendix Section K discussing the limitations of our current method in detail. Importantly, our observation regarding the added complexity of MMDiT's bimodal token streams for block compression should serve as a useful insight for future research.

---

### Official Review · Reviewer_C59U · 2025-11-02

**Soundness:** 3
**Presentation:** 2
**Contribution:** 3
**Rating:** 6
**Confidence:** 5

**Summary:**

The paper proposes a practical recipe for on-device text-to-video generation built on an MMDiT backbone, targeting Qualcomm Hexagon NPU. The framework includes four techniques: 1) Text encoder distillation: replace the original 5B T5-XXL text encoder with a tiny 130M distillT5 + context adapter. 2) Asymmetric VAE decoder distillation with only a 10M parameter model for efficient latent decoding. 3) block pruning for MMDiT by analyzing block importance to reduce parameters for MMDiT, and 4) extend the distribution matching distillation to the pyramidal flow-matching setting.

**Strengths:**

- The paper presents a clear, step-by-step framework for on-device video generation with competitive Vbench.
- Text-only distillation for the text-encoder is well-explained and practically motivated.
- The step-distillation for pyramidal model adaptation is novel and empirically effective under 4‑4‑4 Tab.3, with a dqualitative result of tradeoff.

**Weaknesses:**

- The paper does not include a user study or human evaluation of the generated videos, which would strengthen the perceptual quality claims.
- The hardware/deployment setup is under-specified — key details such as peak memory, per-module latency, data types/quantization, and runtime environment are missing.
- The claim that the visual and textual importance of a block are uncorrelated is interesting, but currently under-supported; additional evidence/analysis would make this argument more convincing.

**Questions:**

- Do you have a latency breakdown for each module of the pipeline?
- Have you conducted a user study or human evaluation to compare the base model with the smaller/pruned model?
- Is there an ablation study showing the effectiveness of block pruning based on importance scores?
- typos:
  - `neogradon` -> `neodragon` in the abstract.

---

> ### Author Response · Authors · 2025-11-20
>
> Thanks for the insightful comments, they have been duly noted. Please find our detailed response as follows:
>
> **User Study:**
> The results of our user study are provided in the Appendix Section I (User studies). We compare three methods: Neodragon (ours), Pyramid-Flow (baseline), and AMD-Hummingbird (external on-device baseline). Consistent with the quantitative VBench scores, users in the study preferred our method in 49.6% of cases, while 29.6% of the cases didn’t report any significant difference between baseline and ours, which also stands as a testament for our optimisations. Please refer to the draft revision for more details of class-specific and comparison to AMD-Hummingbird
>
> **Deployment Setup Details:**
> Please refer to the newly added Sections 4.2 and Appendix Section J, which describe model compilation, on-device hardware measurements, and the quantization details.
>
> **Uncorrelated Visual and Textual Importance:**
> The MMDiT denoiser architecture contains two token streams—one for text and one for visuals—processed through the blocks. As shown in Figure G1, visual tokens exhibit a clear U-shaped importance pattern: the initial and final blocks are critical, while the middle blocks contribute fewer residual details. Textual tokens also follow an overall U-shape, but blocks #13–#18 show a spike in input-output dissimilarity, indicating stronger attention to text tokens.
> Further, as summarised in Figure G4, pruning blocks based solely on text scores or visual scores prevents Stage-1 fine-tuning from recovering the original model’s performance. These observations suggest that textual and visual importance in MMDiT are uncorrelated; otherwise, pruning by either score alone would yield reasonable performance. We have clarified this further in the current revision in Appendix Section G. If you have suggestions for additional probing experiments, we will gladly incorporate them modulo our time and resource constraints.
>
>
> **User Study: Base vs Pruned Model:**
> Due to budget constraints, we could only conduct a single user study (see Appendix Section I). However, the comparison between Neodragon and Pyramid-Flow indirectly addresses the requested study. Moreover, the strong correlation between our user study and VBench scores supports relying on VBench for evaluating individual optimisation effects.
>
> **Typos/Grammatical Errors:**
> We apologise for the oversight and thank you for pointing out; these have been corrected in the current version.

---

### Author Response · Authors · 2025-11-20

We thank the reviewers for their thoughtful and constructive feedback. The overall response to Neodragon has been encouraging, with scores of 8/6/6/4 reflecting strong appreciation for the clarity, practicality, and the rigour of our work. Reviewers appreciated our systematic adaptation of model components for edge deployment (fTZz, eXNS, oTLa), the clear motivation and relevance of on-device video generation for mobile platforms (eXNS), and the well-structured presentation (fTZz). The text-only encoder distillation was praised for practical grounding (C59U), and the novel step-distillation strategy was noted as empirically effective (C59U). Reviewers also found our ablation studies and rich visual results to be comprehensive and convincing (oTLa). We are grateful for this positive reception and look forward to working with the reviewers to further strengthen the paper. Details of the first revision added in individual reviewer responses.

---

### Author Response · Authors · 2025-12-01

Dear AC and reviewers,

We sincerely thank all the reviewers for their encouraging comments as well as their valuable feedback. It is unfortunate what happened with the Openreview leak, but it was completely out of our control.

We wanted to bring your attention towards the review discussion that unfolded before the discussion was interrupted. Our paper originally got the scores of **8** (oTLa), **6** (C59U), **6** (eXNS), and **4** (fTZz); and we incorporated all the suggestions from the reviewers in our revision and provided detailed responses by the suggested deadline of 20th November, well before the interruption. One reviewer (oTLa) maintained their original score of 8 noting our reply and revision.

Although for the other reviewers who didn't provide later comments, we have provided detailed rebuttals and incorporated their suggestions in the revision draft.

We sincerely hope that our analysis and clarifications address all the concerns so far.

Thank you all for your time and consideration.

Best regards,
Authors

---

### Meta-Review · Area_Chair_W6hU · 2026-01-07

**Summary:**

The AC carefully reviewed the paper and the full discussion. The submission received mixed initial scores (6, 4, 6, 8). Reviewers generally recognized that the work successfully adapts a large, server-side Diffusion Transformer into an efficient, deployable mobile solution. They also found the paper well organized, with a data-driven abstract and a clear, logically structured presentation of the method.

However, reviewers raised several concerns. In particular, the reported VBench performance appears to rely on a hybrid pipeline that incorporates external models, and several key compression choices and pruning curricula are justified primarily through empirical evidence. The paper also lacks critical deployment-facing details, including NPU-related metrics and other efficiency measurements, as well as a more objective pruning criterion.

In the rebuttal, the authors addressed the main issues raised during the initial review stage, and the updated aggregated scores now lean toward acceptance. Accordingly, I am inclined to recommend acceptance

**Reviewer Concerns:**

Most of the concerns—about the absence of a user study and under-specified hardware deployment details (Reviewer C59U), the reliance on a hybrid pipeline with external models for the reported VBench score and the lack of theoretical proofs for key compression assumptions (Reviewer fTZz), the unintuitive presentation of efficiency metrics and lack of clear baseline references (Reviewer eXNS), and the unclear experimental settings and missing NPU implementation details (Reviewer oTLa)—were addressed.

**Reviewer Scores:**

I expect all reviewers to maintain their current scores or shift to positive ratings, as the rebuttal and revised manuscript have addressed the major concerns raised during review.

---

### Decision · Program_Chairs · 2026-01-26

Accept (Poster)